# Measuring corporate Paris Compliance using a strict science-based approach

S. Rekker [1✉], M. C. Ives [2], B. Wade [1], L. Webb[1] & C. Greig [3]

The achievement of the Paris Agreement climate goals of well-below 2 degrees of warming requires companies to align their greenhouse gas emission reductions with this goal. To measure whether companies are compliant with the Paris targets we propose several strict conditions that any emissions allocation methodology must meet before it can be classified as Paris-Compliant. Our conditions focus on the need for a common, and early as practicable, base year for all companies and consistency with an underlying Paris-aligned decarbonisation pathway. Additionally, we propose four operationalisation requirements to ensure companies can declare they are on a Paris Compliant Pathway including calculations of their carbon budgets and re-alignment pathways. Applying example Paris-Compliant Pathways and associated metrics to ten high emission electric utility companies and ten cement companies, we find that all but one of these companies are not currently Paris-compliant, with every year of delayed action increasing their required rate of decarbonisation and hence the exposure of billions of investment dollars to transition risk. Applying this proposed method will ensure the Paris carbon budget is met and that progress can be tracked accurately - an imperative for any companies and stakeholders seeking to align their decision-making with the Paris Agreement.

[1] UQ Business School, University of Queensland, Brisbane, QLD, Australia. [2] Smith School of Enterprise and the Environment, University of Oxford, Oxford, UK. [3] Andlinger Centre for Energy and the Environment, Princeton University, Princeton, NJ, USA. ✉email: s.rekker@uq.edu.au

n Paris 2015 international agreement was reached that to avoid the worst effects of climate change global temperature increases must be limited to 'well below 2 °C above pre-industrial levels and pursuing efforts to limit the temperature increase to 1.5 °C'[1]. Stabilising global temperatures in line with this primary objective of the Paris Agreement requires net greenhouse gas emissions to be reduced at unprecedented rates, with $CO_2$ emissions needing to reach net-zero by around mid-century (to stay within 1.5 °C warming levels)[2]. Companies have an important role to play in achieving these goals[3]. They contribute to greenhouse gas emissions directly, through their operations (Scope 1 or 2 emissions), and indirectly, through upstream and downstream emissions (Scope 3), such as those associated with the use of their products[4,5]. Business strategy, research, investments, innovation, and lobbying practices may also have an impact on emissions[6].

Despite evidence of increasing commitment to strengthen global action, most businesses globally are failing to significantly reduce their greenhouse gas emissions[7], an issue exacerbated by a lack of emissions regulations in many countries. Continued business inaction on emissions reductions will render the goals of the Paris Agreement unachievable. Consequences of allowing climate change to continue include considerable economic and financial risk for most companies[8], while evoking ethical concerns for the resulting impacts to the lives of global communities, many disconnected in space and time from the emissions source[9–11].

Addressing climate change requires an increased awareness of climate risk by company executives, the support of regulators, investors, and other stakeholders to ensure firms are managed in a way that they can remain profitable through the transition[12], with a sense of shared responsibility in reducing emissions[13,14]. Fortunately, pressure for companies to take account of their greenhouse gas emissions and associated climate risk has been growing over the last few decades[15–17]. The Task Force on Climate-Related Financial Disclosures (TCFD) has spurred an enormous growth in reporting on climate risk. Most recently, Larry Fink, the CEO of the largest asset manager in the world, BlackRock, in his annual letter to clients asked them to "disclose a business plan aligned with the goal of limiting global warming to well below 2 °C"[18].

For companies to produce an effective business plan for addressing their Paris Compliance requires a consistent and defensible framework for translating this global goal into specific targets for all companies in all sectors[19]. This includes the construction of science-based, fair, and company-specific greenhouse-gas emission metrics that assess a firms' alignment to this mutually agreed climate target.

A number of methodologies for converting global climate targets to company-specific emissions trajectories have been developed[13,20–22]. It is now increasingly common for companies to set climate targets through the Science-Based Targets initiative, which as of August 2021 boasted over 1700 companies "taking action" through this initiative, a tripling in less than two years[23,24]. Emission reduction targets, or specifically, Paris-compliant emission reduction levels, are instrumental in evaluating the performance of companies against these climate goals. Yet, until recently, it was uncommon for science-based methodologies to be used by sustainability rating agencies to measure a firm's climate performance[25].

In this work, we first propose two conditions and four operationalisation requirements to evaluate existing methodologies and assess a company's Paris Compliance (Fig. 1). We find that existing methodologies and organisations fail to meet the minimum requirements we set out in Fig. 1. We demonstrate how Paris Compliance of a company can be assessed, meeting all conditions and requirements. We develop Paris Compliant

emissions reduction pathways, and assess the performance using three metrics, for ten companies each from two very different economic sectors; the data-rich Australian electricity industry, and the data-poor global cement production industry. We find that all but one of the companies are currently not Paris Compliant placing their assets worth billions of dollars at risk for stranding[26] or compromising climate goals.

## Results

### A strict methodology for assessing companies' Paris Compliance.
Translating the goals of the Paris Agreement to the company-specific carbon budgets and Paris Compliant Pathways (PCPs) requires a budget allocation methodology. Unless specified otherwise, references to carbon budgets are company-specific cumulative allocated emissions, so that collectively, companies stay within global carbon budgets. We propose two necessary conditions that must be met for any allocation methodology to be suitable for assessing Paris Compliance, and one desirable condition. First, the underlying global or regional decarbonisation pathway(s) used by the methodology must be consistent with the goals of the Paris Agreement, peaking emissions as soon as possible[27], and holding the increase in the global average temperature to well below 2 °C and pursuing efforts to limit the temperature increase to 1.5 °C above pre-industrial levels[1]. There are multiple pathways that could meet this requirement, and both the choice of pathway and its underlying assumptions must be transparent.

Second, the base year from which progress is measured should be set in 2015 or prior and needs to be consistent with the underlying decarbonisation pathway (of condition 1). We propose a base year of 2015 or prior to reflect the year in which the Paris Agreement was signed and to capture emissions reductions that have been achieved well before 2020. This is also consistent with the "enhanced action prior to 2020" section of the Durban Platform for Enhanced Action CP.21 (2015) referred to in the Paris Agreement[28] and the pathways outlined in the IPCC special report on 1.5 °C[2]. Base years have been shown to clearly have a significant influence on a company's carbon budget[29]. Further, we argue that the same base year needs to be used to fairly compare the performance of companies. Currently, the Science-Based Targets initiative allows companies to choose their own base year. For example, a company could set a base year of 2020 for a target that compares their emissions reductions against a pathway (IEA B2DS) which has a base year of 2014, thus neglecting emissions between 2014 and 2020. If companies are not evaluated against a common, and as early as practicable, base year, it is not possible to compare their actions-to-date against climate goals and it becomes virtually impossible to ensure actions are on target with the Paris goals.

Technically speaking there are and will continue to be global emission reduction pathways that are consistent with a "well-below" 2 °C temperature goal, that start in 2016 or later. For example, the IEA SDS pathway meets our first condition, but has a base year of 2018. However, pathways like these are requiring faster decarbonisation rates with every year of global inaction. More importantly, allowing base years of 2016 or later either allows the global carbon budget to be exceeded or unfairly reduces the established carbon budget of those companies who have been taking appropriate action since (or before) the Paris Agreement was signed - penalising early-movers to compensate laggards. Note that the need for a company's base year to be consistent with the start year of the underlying pathway is a purely mathematical argument – it is required to ensure the world meets the carbon budget of the Paris goals. But requiring that decarbonisation pathways start in 2015 or prior, is an ethical and

| Compliance Conditions | |
|---|---|
| **Required** | **Desired** |
| 1) Underlying decarbonisation pathway consistent with "well-below 2°C" | Differentiated responsibilities |
| 2) Base year consistent with underlying pathway and 2015 or prior | |

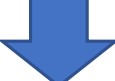

| Operationalisation Requirements |
|---|
| 1) Decarbonisation pathway and carbon budget adjust yearly for changes in actual vs projected company variables (e.g. market share for Sectoral Decarbonisation Approach) |
| 2) Re-alignment decarbonisation pathway is defined that corrects for action deficit, so that the company remains within its carbon budget |
| Additional minor; 3) account for mergers and 4) new companies get time to build up market share |

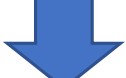

## PARIS COMPLIANT PATHWAY

**Fig. 1 The Paris Compliant company: condition and operationalisation requirements.** This figure summarises the conditions and operationalisation requirements we propose for a companies' decarbonisation pathway to be Paris Compliant, i.e. a Paris Compliant Pathway (PCP).

fairness condition; to keep companies accountable for past emissions, and to ensure companies are compared fairly with their peers. Not accounting for emissions since the beginning of the Paris Agreement makes meeting the Paris goals more and more impractical, and infeasible. Besides, companies are likely to reduce their financial risk by meeting the second condition. Such companies will be exposed to less transition risks than counterparts who delay action, part of the reason some stakeholders want to know whether a company is Paris Compliant.

Finally, a desirable, but not necessary condition for Paris Compliance is that the methodology should account for "common but differentiated responsibilities" in addressing climate change[1]. This means that companies in developed nations have a greater responsibility in mitigating climate change than companies in developing countries due primarily to their greater historic contributions to climate change and mitigation capabilities[1]. Our second condition loosely touches on accounting for historical contributions, but only to 2015 or prior and the underlying pathways may not have distinguished sufficiently

between developed and developing nations. The methodology should be clear on how it allocates the budget amongst companies in different countries.

Applying these conditions, we evaluate four allocation methodologies that have a version of their approach published since 2015: the Sectoral Decarbonisation Approach (SDA)[13,30], Greenhouse gas Emissions per Unit of Value Added (GEVA)[20,30], the Absolute Contraction Approach (ACA)[30] and the Context-based Carbon Metric method developed by the Centre for Sustainable Organisations (CSO)[22] (Table 1). The SDA was the first method adopted and co-developed by the Science-Based Targets initiative in 2015 (an initiative by the Carbon Disclosure Project, World Wide Fund for Nature, UN Global Compact, and World Resources Institute). The SDA takes a sectoral approach using the International Energy Agency's (IEA) decarbonisation pathways[31] and allocates sectoral budgets to companies based on their initial emission intensities, i.e. emissions ($CO_2$) per activity (such as kWh of electricity, or tonnes of steel), and their market share. Initially, companies in "other industries" (for which

**Table 1 The application of our three conditions of Paris Compliance to four allocation methodologies: the Sectoral Decarbonisation Approach (SDA), Greenhouse gas Emissions per Unit of Value Added (GEVA), the Absolute Contraction Approach (ACA) and the Context-based Carbon Metric method developed by the Centre for Sustainable Organisations (CSO).**

| Condition | ACA | GEVA | SDA | CSO |
|---|---|---|---|---|
| 1) Underlying decarbonisation pathway consistent with "well-below 2 °C" | Yes Eligible scenarios selected as per process detailed in SBTi foundations for target setting[27] (An emission scenario envelope derived for well-below 2 °C, and 1.5 °C.) | No Unclear, but 50% GHG reductions between 2010 and 2050. | Yes IEA Beyond 2 °C Scenario (IEA B2DS) — 50% chance of limiting global average temperature rise to 1.75 °C above pre-industrial levels. | Yes - SSP1-1.9 CMIP6; 1.345 °C (CSO, 2021) - CERC-LED-OECD; 1.5 °C. |
| 2) Base year 2015 or prior | No. 2020 (note the SBTi also allows for earlier base years with a higher target for SMEs; i.e. 50% from 2018 in 2035 or 42% from 2020 to 2035). The scenario identified in condition 1 starts from 2020. | Yes 2010 only | Yes Depends on IEA scenario: 2014 for IEA B2DS[31] | Yes 2015[22] |
| 3) Desirable: Differentiated resp. | No Equal absolute reduction target for all companies (4.2%/yr for 1.5 °C and 2.7%/yr for 2 °C); grandfathering | No Allocation depends on economic contributions to global GDP; grandfathering | No Allocation based on initial carbon intensity and market share; grandfathering Minor equity consideration in terms of accounting for different capabilities of sectors and geographical location (geographical location only partially; accounted for in market share, not for intensity) | Yes - SSP 1-1.9 CMIP6 for global. - CERC-LED-OECD for companies in OECD. |

sectoral pathways were not available) were suggested to use the Greenhouse gas Emissions per unit of Value Added (GEVA) method[13], which allocates the carbon budget to companies according to their economic activity. However, the Science-Based Targets initiative has moved away from this approach and is instead suggesting the Absolute Contraction Approach (ACA) for these companies, applying an equal percentage of emission reductions to every company. The Context-based Carbon metric Centre for Sustainable Organisations methodology is similar to the GEVA method, setting targets based on economic activity (contribution to GDP), but updates yearly with actual GDP figures to ensure it remains consistent with a Paris-Compliant carbon budget. It is also important to note that all these methods rely on future company-level projections of physical or economic activity to set targets, which can be different from actual activity.

From Table 1, we find that the SDA and Context-based Carbon Metric by the CSO are the only two methods that meet our two conditions, whilst the CSO also meets the desirable condition of differentiated responsibilities, with companies in developed countries required to decarbonise much faster than those in developing countries.

As several Paris-Compliant pathways will be available for any company to evaluate its performance against, there needs to be transparency in terms of the three conditions of Paris Compliance, i.e. the allocation method used, the base year of the pathway(s) in the underlying method, the scenario this method is consistent with (e.g. IEA B2DS or the SSP1-1.9), and whether they encourage differentiated responsibilities.

**Operationalising a Paris Compliant allocation methodology to derive a Paris Compliant Pathway.** Once a Paris-Compliant methodology is selected and an emission reductions pathway is calculated using this methodology, some additional operationalisation requirements must also be met for a company to claim it is on a Paris-Compliant Pathway. The first operationalisation requirement is that when applying a methodology that relies on the assumption of a future variable (e.g. projected market share, to calculate the emission reduction pathways and

carbon budget), the allocation must be adjusted as soon as the information for the realised variable is available. For example, if market share was an input variable and a projected value was used to calculate the carbon budget in advance, the carbon budget should be adjusted every time the market share differs from the projection.

The second operationalisation requirement is that if a company does not meet the emissions reductions of their PCP in any year, they must construct a "re-alignment" PCP that adheres to all the conditions above while maintaining the company's carbon budget. This will mean that any company which fails to reduce their emissions sufficiently in any year must compensate by increasing their reductions in subsequent years. In short, if a company has not met its target, a "re-alignment" pathway should be defined to make up for the action deficit. In addition, whilst some methods use carbon intensity measures, it is important that corresponding absolute emission pathways and carbon budgets are always calculated[13,29]. Ultimately, the absolute cumulative emissions determine whether the Paris goals are met of holding warming to less than 2 degrees above pre-industrial levels.

Finally, there are two additional minor operationalisation requirements concerning special circumstances. In the case of a merger or acquisition the combined company must reduce its emissions as if both companies have been one company since the base year. In the case of new companies, a defensible approach must also be applied. For example, a new company has five years to establish their market share and emissions, following which they must then align their reductions to a constant reduction rate that achieves net zero as required by the chosen methodology for their sector.

Byway of example, the Science-Based Targets initiative (SBTi) deviates from our compliance conditions and operationalisation requirements in several key areas when operationalising the SDA method. Firstly, the SBTi allows for flexibility in setting the base year, and thus does not account for the emissions of all companies from a common base year that is consistent with the underlying decarbonisation pathway. Second, the SBTi does not strictly require a revision of the target when the company has not met its target or has deviated from its projected market share used

to set the target. The need to account for an "action deficit" is acknowledged in principle by the updated SBTi guidance, with companies that are not on track required to "explain why and the strategy for addressing these deficits in the future"[30]. However, the SBTi still allows companies to set a base year beyond 2014, and even if a company sets a target in 2014 (which is very rare, given companies are encouraged to use the latest year for which is data is available), there is no clarity on how to accurately include any deficits it incurs since this base year, or account for changes in their actual vs projected market share, in their target recalculation.

Accordingly, we demonstrate here our own example Paris-Compliant Pathways (PCP's) that hold true to each of our compliance conditions, and operationalisation requirements, and therefore allow us to demonstrate their importance for ensuring compliance. For this purpose, we create a PCP for a given company, founded on a Paris-Compliant allocation method, in this case the SDA, using the company's initial emissions intensity and projected market (PM) share, or actual market (AM) share where available, from the common base year (2014). This sets the company's Paris-Compliant Pathway for the firm, which has an associated company carbon budget (based on actual market share to date and projected market share). If a company deviates from its PCP in any one year, a "re-alignment" PCP is calculated that would guide the company on how to stay within its carbon budget. In assessing a company's progress, we present the PCP using carbon intensity, $PCP_{intensity}$, as well as using cumulative absolute emissions, $PCP_{emissions}$. Both represent the exact same pathways, but both are useful for tracking Paris Compliance.

Figure 2a, b presents several new emissions intensity pathways for an individual company experiencing a deviation from their PCP. The base year of the PCP is 2014 as this reflects the inception of the IEA B2DS. Our PCP is defined using the company's actual market share for the years to date (2014–2021) and projected market share (here, beyond 2021). This allows us to account for discrepancies between realised emissions and those produced if they followed the base PCP since 2014. Given the company has deviated from its PCP, we illustrate two potential "re-alignment" PCPs that the company could follow to compensate for the lack of progress since 2014 and stay within its company carbon budget. Through this example, we show that decarbonisation rates in the next 5 years have a significant effect on the decarbonisation rates required between 2025 and 2050 if the company is to remain "Paris-Compliant".

**New metrics for gauging companies' transition performance.** Defining conditions and operationalisation requirements for PCPs is not sufficient to evaluate how companies are performing against a PCP. For stakeholders to factor corporate climate performance into their decisions, the extent of their (mis-) alignment with the Paris goals needs to be quantified through metrics. There are two recent initiatives that aim to do this and assess a company's transition performance using the SDA method: 1) The Transition Pathway Initiative (TPI);[32,33] and 2) Assessing low-Carbon Transitions (ACT)[34]. While both initiatives have done an excellent job on engaging with companies on emission reduction pathways, they both inherit shortcomings from the SBTi, with additional issues of their own, which we outline in more detail in the Supplementary Materials. We therefore propose here a set of new metrics to evaluate a company's performance against PCPs.

Having defined what a Paris-Compliant emission reduction pathway is, we can now establish metrics to measure a company's performance against a Paris-Compliant Pathway. We propose three new metrics, mostly focused on absolute emissions ($PCP_{emissions}$), for evaluating the Paris Compliance of companies.

"Metric 1 - Performance to Date" measures current performance since the base year, "Metric 2 – Projected Performance" provides indicators for the implications of continuing business as usual, and "Metric 3 - Re-alignment Decarbonisation Rate" gives the re-alignment factor required for the company to return to a climate-safe pathway. For measuring Metric 1 - Performance to date we compare a company's cumulative emissions to date against its $PCP_{emissions}$, where $PCP_{emissions}$ are defined as the cumulative emissions that would have been emitted since 2014 had it followed its $PCP_{emissions}$. For Metric 2 – Projected Performance we estimate a reference emission pathway based on a projection, and estimate three sub-metrics: 2a) the "Estimated Year to Finish" (EYF), the year the company carbon budget would be exceeded, 2b) their projected (carbon positive) production in the EYF, and 2c) the exceedance of the company carbon budget as a multiple in the year the company's PCP must become net-zero (around 2050 for the electric utilities sector in the IEA B2DS scenario). The carbon budget is defined as the total emissions a company can emit before it must achieve net-negative carbon intensities (the International Energy Agency B2DS allows for temporary overshoot of the carbon budget). For Metric 3 we quantify two re-alignment options that provide a measure of how much faster a company will need to decarbonise relative to the rate in its original $PCP_{intensity}$.

Figure 2 demonstrates our three metrics for AGL, an Australian electricity provider. The actual $CO_2$ emissions (dark blue line) of the company since 2014 exceeded their Climate-safe Emission Pathway (green line) by a factor of 1.20 in 2021. Under a "maximum action" scenario, the company would exceed its carbon budget in about a decade (2030) (metric 2a), whilst still producing 33.3 million MWh (with positive carbon intensities). Thus, the company will need to retire this production capacity early (or employ eligible $CO_2$ capture and storage and/or negative emission technologies) if it wants to remain within its carbon budget. If they continue on this projected pathway their emissions will be 144% of the company's carbon budget by 2050 (Metric 2c). Note that for other sectors with limited data availability, we can project emissions using, for example, the geometric growth of emission intensity since 2014, which we have done for our sample of cement companies in Fig. 3d–f. To avoid having to apply such crude measures companies themselves are encouraged to release projections of their emission pathways.

If the company takes immediate action (in this case, in 2022) to re-align its trajectory to stay within its carbon budget to 2051 (the year the company should go carbon negative to follow its PCP), it will need to decarbonise its operations 1.77 times more rapidly than had it followed its PCP since the base year of 2014 (Table 2, Fig. 2). If the company delays re-alignment and continues its recent (2014–2021) decarbonisation pathway until 2027, it will need to decarbonise 4.14 times more rapidly post-2025 than if instead it had followed its PCP since 2014. This demonstrates the impact of delayed action especially in the early years, as inaction can lead to unachievable rates of decarbonisation. This increased pace of change will be highly disruptive, forcing the company to accelerate the retirement of carbon-intensive assets and more rapidly mobilise capital for low-carbon assets. Earlier emission reductions would allow for a slightly more gradual decarbonisation. These results are summarised in Table 2.

Figure 3 demonstrates the application of the metrics to the ten largest Australian electricity utilities and ten largest cement companies from various countries, showing the scale of the decarbonisation challenge for these sectors. For our electric utility sample, we have actual asset data and can estimate a "maximum action" scenario where the company continues to operate its assets until the expected closure date upon which the assets are replaced with zero-carbon alternatives, and market share is kept

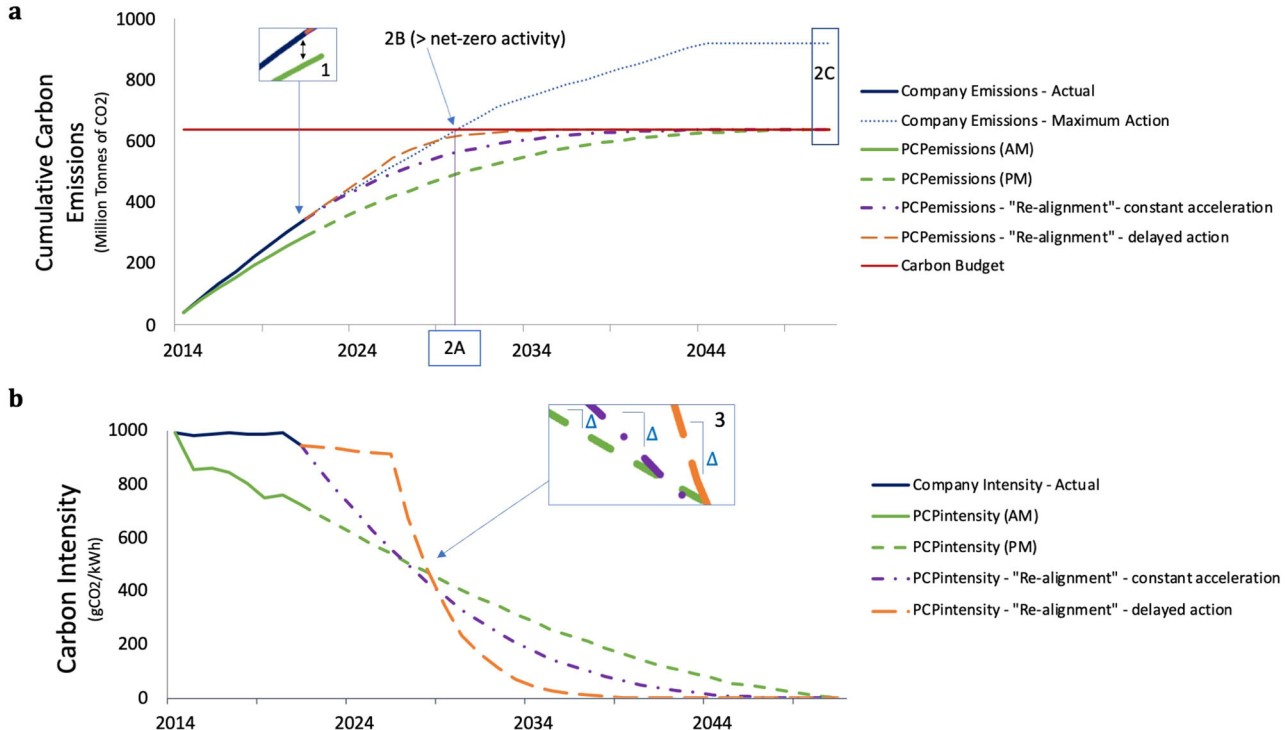

**Fig. 2 Paris Compliant Pathways and transition performance metrics for the largest Australian electric utility company (AGL).** Both Panel **a** and Panel **b** show the same Paris Compliant Pathway (PCP) using the Sectoral Decarbonisation Approach on an existing company's initial intensity (2014) and actual market share (green line) and projected market share (dashed green line), where Panel **a** shows carbon intensities and Panel **b** shows cumulative carbon emissions. It also shows two possible "re-alignment" PCP's the company could follow to stay within its carbon budget: a PCP featuring a constant "accelerated" decarbonisation rate which commences immediately (dotted yellow line); and a PCP which follows recent (2014–2021) decarbonisation rates for the next 5 years, followed by a new accelerated decarbonisation rate (long-dashed orange line). Panel **a** shows Metric 1 and metric 2. Metric 1 measures the performance since the base year (cumulative emissions since the base year 2014 relative to the Climate-safe Emissions Pathway). Metric 2 is composed of three sub-metrics; 2A is the estimated year that the company carbon budget will be fully emitted based on a "maximum action" scenario of future cumulative emission projections, all assuming constant market-share (short dash). A "maximum action" scenario is based on the company's most recently announced closure dates for fossil fuel generating assets and assumes the replacement of such assets with zero-emissions generation capacity and their market share being kept constant; 2B is the amount of activity in place under the "maximum action" scenario at the EYF with a positive emission intensity; 2C is the emissions under the "maximum action" scenario in 2050 compared to the carbon budget. Panel **b** also illustrates Metric 3 which measures the difference in decarbonisation rates between the PCP$_{intensity}$ and the "re-alignment" PCP$_{intensity}$.

constant. In 2021, all but one Australian electric utility company exceeded their PCP (Fig. 2a), and if the companies followed a "maximum action" scenario from 2021 onwards, all but two companies are expected to exceed their total carbon budget before 2035 (Fig. 2b) and exceed their carbon budget by between 1.2 and 3 times in the year 2050 (Supplementary Data Fig. 4). The company Engie retired all its Australian coal-fired power stations at the start of 2017, which consisted of more than 70% of its total generation in 2016. This has led to the company being Paris Compliant (score on metric one is less than 1), but also to never exceed its carbon budget under the "maximum action" scenario, as their remaining gas plants will all close by the end of 2037 and are assumed to be replaced with renewables. Similarly, even though Origin is currently not on track, they are not expected to exceed their carbon budget given their carbon-intensive assets are expected to close prior to reaching the carbon budget. The cement companies are slightly more aligned with their PCP, yet all ten companies have exceeded their PCP to date (Fig. 2d) and are expected to exceed their carbon budget by 2046 (Fig. 2e).

As a result of their failure to follow their PCP, all but two electric utility companies, and all cement companies, in our sample must increase their rates of decarbonisation between 1.01 and 6.6 times faster than required had they followed their PCP (Metric 3; Fig. 2c, f). This metric is a measure of the extent to which a firm's present decarbonisation trajectory is misaligned

with the Paris Agreement, as defined by the PCP. Any further exceedance of the PCP beyond the last reported year will increase this rate even further.

We used the SDA method to demonstrate the PCP approach. It should be noted that the SDA method itself also has some known limitations within its foundations[35], such as allowing companies with a higher initial intensity to have a greater share of the remaining global carbon budget (i.e. grandfathering)[13]. In addition, the IEA B2DS pathway used by the SDA relies on significant amounts of carbon capture and storage and negative emissions in the second half of the century. This poses two problems: first it implies a significant overshoot of the carbon budget during the first half of the century followed by significant drawdown; and second, it relies on the rapid later expansion of technologies which are currently not being deployed significantly at scale[31,36]. This may not alter the total carbon budget but could have implications for the sectoral breakup of the budget. The B2DS pathway is also only consistent with the more conservative interpretation of the Paris Agreement with a 50% chance of keeping global warming below 1.75 °C. Further, company data availability is a concern in calculating PCP's, especially for the SDA method as it requires production output in addition to emissions. We deliberately demonstrated PCPs with a data rich set of companies (Australian energy providers) and a data poor set (global cement companies) to demonstrate PCPs could still be

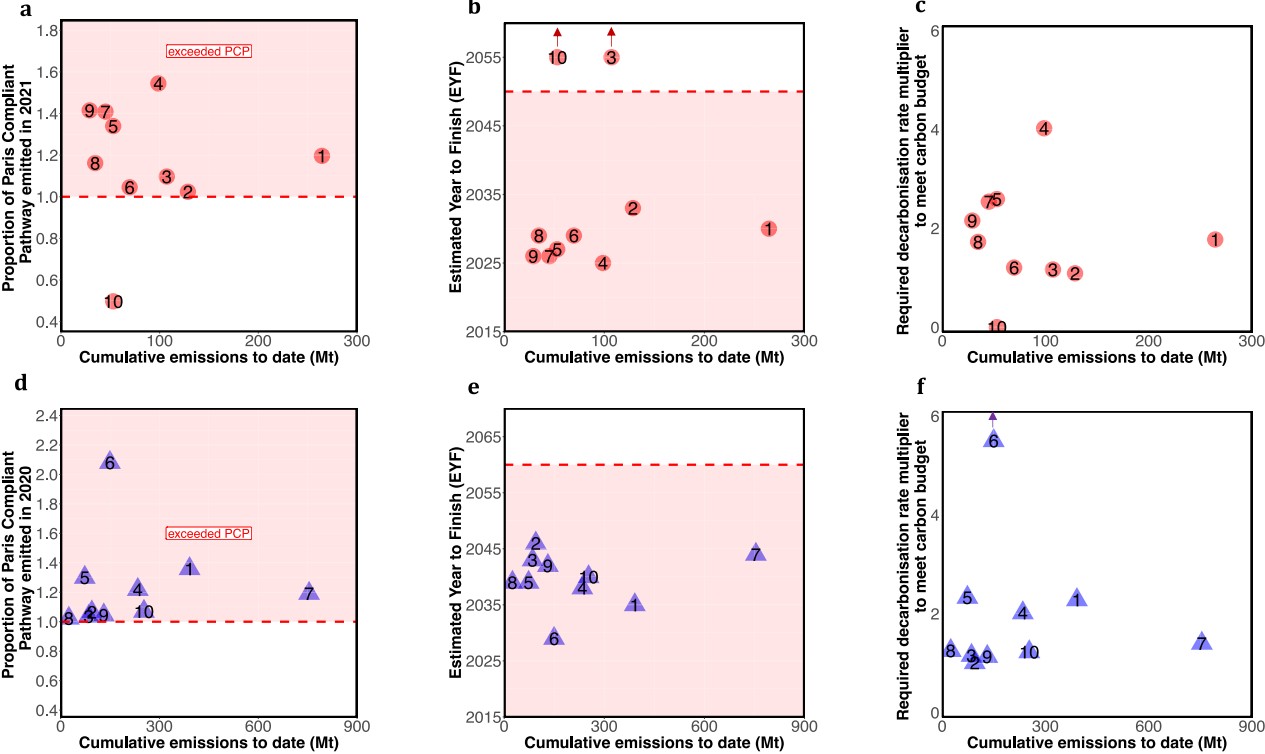

**Fig. 3 Transition performance metrics for the ten largest producing electricity generators in Australia (a–c, red circles), and ten cement producers from various countries (d–f, blue triangles) for metric 1, 2a and 3.** Panels **a** and **d** show Metric 1 - the proportion of the Paris-Compliant Pathway (PCP) (in absolute emissions, PCP$_{emissions}$) actually emitted in 2021 since the base year 2014, the shading above a value of one indicates the company has exceeded its PCP. Panels **b** and **e** show Metric 2A - the Estimated Year to Finish the carbon budget if carbon intensity is decarbonised according to the "maximum action" scenario (panel **b**) or geometric carbon intensity growth 2014–2020 (panel **e**), the shading below 2050 for Electric Utilities (panel **b**) and 2060 for Cement (panel **e**) indicates the company is expected to finish its entire carbon budget prior to the timeframe given by the underlying decarbonisation pathway (IEA B2DS). Panels **c** and **f** show Metric 3 - the multiplier on decarbonisation rates required (compared to if it had followed its PCP (in carbon intensity, PCP$_{intensity}$) since 2014) to be Paris-Compliant. Additional metrics are shown in Supplementary Data Fig. 4. Panels **a–c**: 1 = AGL, 2 = Energy Australia, 3 = Origin, 4 = Stanwell, 5 = CS Energy, 6 = Alinta, 7 = Delta, 8 = Millmerran, 9 = Callide and 10 = Engie. Panels **d–f**, 1 = Heidelberg cement, 2 = ACC, 3 = Ambuja cement, 4 = Ultratech cement, 5 = Shree cement, 6 = CRH (LON), 7 = Holcim, 8 = Asia Cement, 9 = Siam Cement, 10 = Cemex CPO.

**Table 2 Calculated transition performance metrics for company in Fig. 2.**

| Metric | | Value for company | |
|---|---|---|---|
| 1: Cumulative Emissions emitted since base year relative to emissions allowed under the Climate-safe Emission Pathway (%/absolute) | | 120%/57MtCO$_2$ | |
| 2: The potential for stranding of existing assets (using a "maximum action" scenario) | | | |
| A: Estimated Year to Finish (EYF) Carbon Budget | | 2030 | |
| B: Production Activity with greater than net-zero emission intensity in EYF | | 33.3 million MWh | |
| C: The level of exceedance of the company's carbon budget by 2050 (percentage of company's carbon budget) | | 144% | |
| 3: Adjusted decarbonisation rate required to be climate-safe (compared to rate if followed PCP$_{intensity}$) | | 2022–2026 | 2027–2051 |
| i) constant acceleration | | 1.77× | 1.77× |
| ii) delayed action (2014–2021 decarbonisation rate in next 5 years) | | 0.12× | 4.14× |

adequately applied. Furthermore, the companies themselves can still make the necessary calculations to provide stakeholders interested in their Paris Compliance, and our PCP approach provides directions to stakeholders on what data to request from companies.

## Discussion

The PCP approach provides companies with a means of accurately ensuring their efforts are aligned with the goals of the Paris Agreement. Our three proposed metrics can thus be used by companies, investors, and other stakeholders to evaluate the

company's transition performance. A company's transition performance is related to their exposure to transition risk[26,37–39] which manifests primarily as assets stranded due to a reduction in demand for their services, from technological change, or from regulatory, legal, and fiscal measures to reduce emissions[37,40]. The more a company exceeds their PCP, the larger the company's misalignment with meeting the Paris goals, and the more likely they will be exposed to such transition risks. However, it should be noted that the concept of transition risk is much broader than stranded assets and can manifest as a loss of market share (market risk), as a disadvantage through ownership of inferior

technologies (technology risk), exposure to litigation (legal risk), and erosion of investor or consumer confidence (reputation risk)[39]. As some of these risks involve historic emissions and actions (e.g. legal and reputation risk) they are not fully covered by the PCP metrics.

Nobel Laureate Milton Friedman once famously wrote that the social responsibility of business was to increase its profits[41]. However, society is now putting much loftier demands on companies, and the many stakeholders now have much more influence on corporate behaviour, such as shareholders, most obviously through the exclusion or inclusion of companies' debt or equity in their portfolios[42–44]; regulators, central banks, and banks, through lending restrictions and disclosure requirements[45]; and customers, suppliers, employees, politicians, academics, and communities through purchasing decisions, employment preferences, publications, policies, and various forms of activism[46–50]. The failure to accurately monitor, verify and benchmark corporations' performance on GHG emissions obscures the perceived risks for these stakeholders[51] and limits their capacity to make decisions aligned with the Paris Agreement. If companies can claim "compliance" while exceeding carbon budgets, then the Paris goals will not be achieved, and investors will have been misled.

We have presented here a strict science-based methodology for measuring companies' compliance with the goals of the Paris Agreement. We have specified the exact methodological conditions and operationalisation requirements for producing Paris-Compliant Pathways consistent with the goal of achieving a "well below 2 °C" transition to net-zero emissions. We have also provided three metrics for evaluating company's performance against their pathways and demonstrated how such metrics can be used to evaluate companies within different sectors. When applied to major corporations in both the Australian electric utility sector and the global cement sector our metrics showed alarmingly that all companies were clearly not on track to meet the goals of the Paris Agreement, and that their decarbonisation efforts will need to be ramped up significantly to avoid increased exposure to transition risk.

On a cautionary note, while our method holds companies more accountable to decarbonisation commitments, it is not without its limitations, such as the limited number of modelled pathways that meet our strict criteria. Companies might reasonably prefer to align their PCP with a modelled pathway other than IEA B2DS. Given the plethora of macro-scale energy transition and integrated assessment models (IAMs) available, it may well be that specific models are perceived to offer a higher level of resolution and/or more up-to-date assumptions for their sector and region. However, with different companies in different sectors variously aligning themselves with different decarbonisation pathways, it is likely that the aggregated company carbon budgets would be quite different to the budget if all companies were aligned to a common path. This is especially problematic because of the number of IAM scenarios that allow significant overshoot of carbon budget followed engineered drawdown of $CO_2$ later in the century. This overshoot issue is particularly troubling, and one that deserves more attention by researchers and institutions considering the alignment of firms and sectors with the Paris Agreement.

For now, our proposed conditions and requirements for Paris Compliance provide a consistent approach for calculating and comparing the company's Paris Compliance and transition performance. Regulators, investors, customers, researchers and other stakeholders must also play their part in ensuring that the firms that correctly report and demonstrate Paris Compliance are rewarded with profitability through their transition[12,52].

## Methods

**Paris-Compliant Pathway (PCP)**. We use the Sectoral Decarbonisation Approach (SDA) initially published by Krabbe et al. (2015)[13] and updated by the Science-Based Targets initiative (2019) to determine a companies' Paris-Compliant Pathway (PCP). The SDA methodology sets carbon intensity targets based on a company's initial carbon intensity, the IEA B2DS' sectoral decarbonisation pathway and their projected market share. The IEA B2DS' pathway is consistent with a 50% chance of limiting average future temperature increases to 1.75 °C. The PCP_{emissions} is a pathway of cumulative emissions that would be consistent with an IEA B2DS pathway provided all companies stay on or below their PCP.

The $PCP_{emissions}$ is calculated as follows:

Paris-Compliant Pathway ($PCP_{emissions}$) for a company in year y (base year 2014):

$$PCPemissions_y = \sum_{y=2014}^{y} PCPintensity_{y,SDA} * CA_y \qquad (1)$$

Where the $PCP_{intensity}$ in year y is calculated using the SDA methodology:

$$PCPintensity_{y,SDA} = dp_y m_y + SI_{2060} \qquad (2)$$

Where

$$d = CI_b - SI_{2060},$$

$$p_y = \frac{(SI_y - SI_{2060})}{(SI_b - SI_{2060})},$$

$$m_y = \frac{(CA_b/SA_b)}{(CA_y/SA_y)}$$

$CI_b$ = Carbon intensity in the base year (2014) set by IEA B2DS

$SI_y$ = Sectoral Intensity in year y set by IEA B2DS

$SI_b$ = Sectoral Intensity in base year (2014) set by IEA B2DS

$SI_{2060}$ = Sectoral Intensity in 2060 set by IEA B2DS

$CA_b$ = Company Activity in base year (2014)

$CA_y$ = Company Activity in year y

$SA_b$ = Sectoral Activity in base year (2014)

$CA_y$ = Sectoral Activity in year y

Sectoral activity (SA) is set by the IEA B2DS pathway, and we use the activity for the region the company produces (so we use IEA B2DS-consistent activity for the OECD for Australia).

We distinguish between a) a PCP_{intensity} calculated using actual market share (AM) where available, thus in 2021 for electric utility companies and 2020 for cement companies the PCP_{intensity} is based on actual market share 2014–2021 and 2014–2020 respectively, and b) a PCP_{intensity} calculated using projected market share (PM) for future years, assuming the company had followed their PCP_{intensity} to date.

Whilst a company can set targets for their PCP using their projected market share, the PCP can only be finalised once the activity, and thus market share, of a company is known (usually at the end of the year). Therefore, the PCP needs to be continuously revised when new market share data becomes available. We can therefore distinguish between two PCPs: actual and projected. For the actual PCP, $m_y$ and $CA_y$ is known, for the projected PCP, $m_y$ and $CA_y$ is projected, keeping market share constant according to sectoral activity projections (OECD or non-OECD). For the year $t$, this results in:

$$Actual\,PCPemissions_t = \sum_{y=2014}^{t} PCPactualintensity_{y,SDA} * CA_y \qquad (3)$$

$$\begin{aligned} Projected\,PCPemissions_{t+n} = {} & Actual\,PCPemissions_t \\ & + \sum_{y=t}^{t+n} PCPprojectedintensity_{y,SDA} * CAprojected_y \end{aligned} \qquad (4)$$

**Metric 1**. To measure performance to date, we compare their actual cumulative emissions to date, i.e. their actual "Emission Pathway", to the Actual PCP and calculate the percentage of PCP emitted to date. We calculate the actual EP as the company's actual carbon intensity multiplied by the company's actual activity.

$$Actual\,EP_t = \sum_{y=2014}^{t} CI_y * CA_y \qquad (5)$$

We can then calculate the Corporate Climate Performance to date:

$$Corporate\ Climate\ Performance\ to\ date = \frac{Actual\ EP_t}{Actual\ PCP\ emissions_t} \quad (6)$$

If >1, the company has emitted more than their $PCP_{emissions}$ (not consistent with Paris goals)

If ≤1, the company has emitted the same or less than their $PCP_{emissions}$ (consistent with Paris goals)

It is important to note that we use $PCP_{emissions}$ and not $PCP_{intensity}$ here. The $PCP_{intensity}$ simply captures what the intensity should be in a certain year but gives us no indication on whether a company has exceeded its $PCP_{intensity}$ to date, and by how many tons of emissions. This is captured by $PCP_{emissions}$.

**Metric 2**. To understand the implications of a company's performance to date and their projected emissions, we use three sub-indicators (metric 2a, 2b and 2c). We project emissions forward for the electric utility companies using a "maximum action" scenario. In this scenario, the company produces until closure date and replaces its plant solely with net-zero carbon (e.g. renewables, nuclear or gas + CCS). We understand that this trajectory cannot easily be used for other sectors, where plant/asset data and intensity are not available. Also, ideally real company projections are used to estimate these trajectories. We projected the plant activity and emission intensity to remain constant at the 2019–2021 average activity. However, we assumed overall activity would grow in line with the OECD activity for the sector. If granular data on projected activity is not available, we suggest projecting forward using other estimation methods proposed in Rekker et al. (2018), such as recent emission growth or reduction rates. For the cement companies in Fig. 3d–f we have projected forward using the 2014–2020 Carbon Intensity geometric growth/reduction rate.

Metric 2a follows Rekker et al. (2018)[20] by using the "Estimated Year to Finish", which is the year the company is estimated to have emitted their full company carbon budget. We define the company's carbon budget as the total amount of carbon the company can emit to operate consistently with the chosen climate pathway (e.g. IEA B2DS) in the respective timeframe (e.g. 2014–2060 for IEA B2DS). Under the IEA B2DS however, and the power sector specifically, there is a temporary overshoot allowed. The Sector Intensity for power goes negative from about 2050 to 2060. Therefore, we define the carbon budget of a company at the year that the $PCP_{emissions}$ is at its highest level:

$$Company\ Carbon\ Budget = \max(PCP) \quad (7)$$

This is subject to the company generating negative emissions after the year y that satisfies max (PCP) such that it reaches $PCP_{2060}$ in the year 2060.

This is slightly different to the approach of the SBTi, who uses 2050 as the year of convergence for all the sectors. Note that the company carbon budget is not fixed, it depends highly on the projected company activity (captured under *Projected PCP_{emissions}*). It should be updated and refined on a yearly basis. Assuming the company has not yet exceeded its carbon budget in 2021, the remaining carbon budget in year "EYF" can be calculated by subtracting the emissions to date (in year y) from the Carbon Budget:

$$Remaining\ Carbon\ Budget_y = Company\ Carbon\ Budget - Actual\ EP_y \quad (8)$$

The firm is estimated to finish its remaining carbon budget in the year "EYF" that satisfies:

$$Actual\ EP_{EYF} = Company\ Carbon\ Budget \quad (9)$$

The EYF is based on a lenient company carbon budget (the company will have to generate negative emissions after), thus the EYF is also a lenient measure.

Metric 2b measures the production levels ($CA_y$) that are estimated to be still in place when the company reaches its EYF (metric 2a). This is a measure of how much energy generation capacity will need to be retired at the EYF if the company is to continue with its projected outputs but stay within its carbon budget. Metric 2b is observed from the difference in the estimated emissions in the EYF and the year prior.

Metric 2c evaluates how many times the company's carbon budget will be exceeded if the company continues along a "maximum action" scenario when their $PCP_{intensity}$ should reach zero, which is around 2050 (but depends on the individual company). It is calculated as the cumulative emissions in the year the $PCP_{intensity}$ should reach zero, divided by the company's carbon budget.

**Metric 3**. To stay within the carbon budget consistent with the IEA B2DS' pathway, companies that have emitted more than their fair share (i.e. their total $PCP_{emissions}$), will have to compensate for this in the future. This is closely aligned with transition risk– the companies that have decarbonised the least to date are the most vulnerable to a transition to a world that meets the Paris goals.

To understand the pathway a company should follow to re-align its emissions, we calculate a "re-alignment factor". This factor captures how much faster the company must decarbonise in order to maintain its projected outputs and stay within its carbon budget, compared to when it had followed the $PCP_{emissions}$ from the start.

To calculate this re-alignment factor, we first calculate the speed of decarbonisation required if the company had followed $PCP_{emissions}$ since 2014. We derive this 'speed' by calculating the $PCP_{intensity}$ for the firm, and then inferring

the required annual decarbonisation rates to adhere to this pathway. With the acquired decarbonisation rates, we can computationally solve the following equation simultaneously to solve for the (constant) z value using goal seek in excel.

$$Re - alignment\ CI_{t+1} = CI_t \times (1 + z \times d_t) \quad (10)$$

Such that

$$\sum_{n=2021}^{2050} Re - alignment\ CI_t \times CA_t = Remaining\ Company\ Carbon\ Budget \quad (11)$$

**Data process**. Our data was collected and processed in Microsoft Excel (version 16.59). To select our sample, we sourced the activity data for all Australian electric utility companies on a disaggregated level (generator-by-generator activity from NEM dispatch data provided by the AEMO) and aggregated the data to a firm-level. We used the NEMReview6 software[53] and sorted each generator by owner, then calculated the total activity for each firm by summing the generators' activity for a given time interval. We did this for the years 2014–2021.

We ranked the firms by their activity in 2014 and selected the 10 largest producing firms for our sample. However, due to the nature of some of the corporate entities, certain firms had to be excluded. One firm, Hydro Tasmania, was excluded from the analysis as this firms is a renewable energy generator. Due to the hydro power resulting in much lower carbon intensities[54], their inclusion would have complicated the method to construct a Carbon Budget, and result in these firms having a negative carbon budget (i.e. the method expects these firms to withdraw emissions from the atmosphere that are not related to their own production).

To source company-specific emission data, we first considered the National Greenhouse and Energy Reporting (NGER) database published by the Clean Energy Regulator (CER). Since 2007, Australian corporations have been required by law to report their greenhouse gas emissions to CER each year. Although this may seem the most logical (and most accurate) dataset for company-specific emissions, this data source was not fit for our analysis for the following reasons: i) The data is reported in financial years, whereas the SDA method is constructed using calendar years. As the NGER data is published as an aggregate, there was/is no possibility of transforming the data from financial years to calendar years, ii) In the earlier reporting years, certain firms chose to publish emissions by holding company (e.g. Loy Yang, Tarong) rather than consolidating it under the 'final' owner (e.g. AGL). To avoid this complexity, we use the same method as we used for the activity data, and thus sourced disaggregated emissions data by generator (using the NEMReview 6 software) and aggregate it to a company level.

We retrieved the IEA B2DS sectoral data from the International Energy Agency, Energy Technology Perspectives 2017 - www.iea.org/etp2017.

To source the data from cement companies, we used the Thomson Reuters Datastream (Refinitiv) service[55]. We retrieved emission data (scope 1 and 2) and carbon intensities (scope 1 and 2 per tonne of cement), from which we derived cement production in tonnes of cement. From this we could calculate scope 1 emission intensities. We had complete data from 2014 to 2020 for eleven companies, and selected the largest ten companies based on 2014 activity levels.

**Reporting summary**. Further information on research design is available in the Nature Research Reporting Summary linked to this article.

## Data availability
The company scores for Figs. 2 and 3 are provided in the Supplementary Materials. The underlying data and the calculations are available under restricted access (repository https://doi.org/10.48610/289d707) given the emissions and production data are obtained through purchased licenses or free trials. For the electric utilities data we retrieved emission ("Greenhouse gas emissions") and electricity generation ("Metered Generation (As Generated)") data from the NEM6Review software (see https://v6.nemreview.info/what/buy/step1.aspx). We used the Refinitiv (Thomson Reuters Datastream) data services to obtain emissions and production data for the cement companies.

## Code availability
The figures were created using R. The R code used to generate the figures is available in the Supplementary Data.

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

## Acknowledgements

We would like to thank Rupert Stuart-Smith and Richard Heede for their valuable reviews on earlier versions of the manuscript.

## Author contributions

S.R., B.W. and C.G. designed the study; S.R. collected, verified, processed and analysed the data; S.R., M.C.I., B.W, and C.G. contributed to the writing of the article, the methods, interpretation of the results, and article revisions; L.W. assisted with data collection and analysis.

## Competing interests

The authors declare no competing interests.
