## [Peer Review File · Nature Communications]

Reviewer comments, first round

Reviewer #1 (Remarks to the Author):

This paper presents new concepts and metrics related to measuring companies' CO2 emission performance against "Paris compliant" benchmarks. The authors do a relatively good job at describing the weaknesses of existing approaches (TPI and ACT) and present their own approach in a clear way using a case study. The paper makes a significant contribution to the emerging literature that address the question "what does a Paris-compliant company look like?". The paper is certainly relevant to several academic fields and corporate stakeholders (regulators, investors, etc.).

The only significant weakness of the paper, from my perspective, is the presentation of the SDA method and its use by companies in the introduction section, where a number of inaccurate claims are made (see my detailed comments below). I also think the clarity of the text presenting the limitations of SDA, TPI and ACT (around page 4-7) can be improved (see my detailed comments below). In addition, I have a list of general and specific comments that mostly relate to increasing the readability of the paper:

General:

- Your text contains a large number of abbreviations, especially in the methods section. Consider getting rid of some of the abbreviations used less frequently, to increase readability.
- Consider using more informative names than "Metric 1/2/3".
- Why is the electricity provider in Fig. 1 and 2 anonymous, given that the ten providers in Fig. 3 are disclosed? Why not be transparent about all case companies used throughout the study?

Specific:

- Line 39: More prescriptive than what? Please clarify.
- Line 70-71: I am not sure I understand the reference to the footprint concept here. Consider rewording.
- Line 72: You may also want to refer to the Context-based carbon metric (for businesses) from CSO: <https://www.sustainableorganizations.org/context-based-metrics-public-domain/>. The spreadsheet tool actually contains a function that allows users to adjust emission pathways (in order to conserve the originally developed cumulative emission target) along the lines that you suggest in your paper.
- Line 73: Can you provide a reference in support of your statement that "the Sectoral Decarbonisation Approach (SDA) is the most widely adopted"? From what I know, only a minority of companies with an SBT approved by the SBTi disclose the method that they used to set the target. In addition, I suspect that most companies with recently approved SBTs have used the ACA method, since it is applicable with 1.5 degree emission pathways and to all sectors, whereas the SDA method is currently only applicable with less ambitious emission pathways ("2 degrees" and "well-below 2 degrees") and to a handful of "homogenous" sectors.
- Line 78: For reasons given in my comment above, the statement "it can be applied to any sector" is not true. The main difference between SDA and other SBT methods is that it allows calculation of sector-specific targets, based on the sector-specific emission pathways of the IEA. But since the IEA emission pathways do not cover all sectors of the economy (agriculture is left out, for example), the SDA method cannot calculate targets for all sectors. By comparison, all other SBT methods are applicable to all sectors.
- Line 79: I am not sure that I have seen "market share" and "initial carbon intensities" recognized as "equity principles" in the literature. Can you provide a reference? You may also consider that the SDA method relies on the principles Grandfathering (the higher the initial absolute emissions, the higher the future allowed absolute emissions), Convergence (the carbon intensity of all companies are assumed to converge in 2050) and Cost-optimization (the IEA sectoral pathways have been derived by considering differences in the costs of reducing emissions across sectors).
- Line 99-115: It is unclear whether the three "key conditions" are presented by the SDA developers/SBTi or by you. For example, the SBTi does not, to my knowledge, require all companies to use 2014 as baseline year (as you note below). Please clarify the origin of the three

“key conditions”.

- Line 132: Do you mean “the updated SBTi guidance”?
- Line 135: “even if a company sets a target in 2014, there is no clarity on how to accurately include any past deficits” This is true. But SBTs are always derived from performance (emissions and activity) in a single baseline year and, as such, are independent of performance prior to that year. Given this scope, it is not surprising that SBTi does not provide clarity around “past deficits” What does “past deficits” even mean in this context? Consider rewording.
- Line 144: Revised in response to what? Emissions not following the originally targeted emission trajectory? Please clarify. Also, from Table 1 it appears that TRI does include revised pathways. Please ensure consistency between text and table.
- Line 152: Consider elaborating that this means that the ACT approach is only consistent with the least conservative interpretation of the Paris Agreement (preventing global warming of more than 2 degrees).
- Line 157: “a company’s original allocated carbon budget would be exceeded” - I think that this is the central limitation of ACT (and TPI). Perhaps you could make this clearer from Table 1. I think the point relates to your benchmark about “action deficit”, but if you were to adopt wording related to “carbon budget exceedance” (or something along those lines) you might more effectively convey your message.
- Line 203: I am confused about the last part of the sentence – “that produces a greater than zero emission intensity”. Would there be any cases where the projected production capacity in the EYF would lead to negative emission intensities? Consider rewording.
- Line 206: “the IEA B2DS allows for temporary overshoot” – overshoot of what? The underlying temperature goal? Please specify.
- Line 207: Should be “as shown in Figure 1c”, right?
- Line 212: Are you sure there should be a “hence”? I do not see the link between requiring only publicly available data and using a simple binary carbon performance metric. For example, why does the use of publically available data prevent a more quantitative assessment that goes beyond aligned/not aligned labels?
- Line 228: “metric C” should be “metric 2C”, right?
- Fig 2.: It is a bit confusing that the “ACT – Current benchmark” has a different colour than in Fig. 1 (in which a similar red/brown colour is used for “ACT – Previous benchmark”). Please align, if possible.
- Line 249: Please remind the reader what year “immediate” refers to.
- Line 249-253: These results are shown graphically in Figure 2b, right? If so, consider referring to the figure here.
- Line 265: “Overall results are very similar to if companies followed a “maximum action case””. Starting in 2014? Please specify. The point also applies to the next sentence.
- Line 286: “The additional metrics” - Are you referring to Metrics 1, 2 and 3? Please clarify.
- Line 452: I see from the equation that you assume convergence of company emission intensity pathways in 2060 (for a given sector). This is consistent with what the SBTi used to assume. However, since their latest version of the SDA tool, incorporated in the broader SBTi tool from June 2020 (version 1.2 - <https://sciencebasedtargets.org/resources/files/SBT-Tool-v1.2.1.xlsx>), convergence is assumed to happen in 2050 (as originally assumed in Krabbe et al.). Therefore, you might want to either update your equations and results (which would not change much, I expect) or, at least, make a note about the year of convergence currently assumed by the SBTi.
- Line 481: If I understand correctly, you propose to use “regional sectoral activity projections” and assume a constant market share for each company within? This raises two questions: 1) What is the data source for these “regional sectoral activity projections”? - I do not see this mentioned in your Data section. 2) Why not use company-specific projections, if available? – Such projections may either be made by companies themselves or external analysts and are usually available in financial databases, such as Bloomberg and FactSet, though usually only extending around 3 years into the future (which is a big limitation in this context, I know). I encourage you to address these questions in the Methods text.
- Line 542: I am not sure that I understand what “The metric” refers to. The previous sentence is about energy generation capacity, but this sentence is about emission trajectories, right? Consider rewording.
- Line 600: Does this also mean that the SDA method is not applicable to hydro power companies? Consider making a note about this.

Reviewer #2 (Remarks to the Author):

This paper contributes to an emerging literature on assessing corporate climate action. It is an important topic, and more academic scholarship and constructive debate is urgently needed, particularly since a lot of research is published in the grey literature rather than in peer-reviewed academic journals.

This paper builds on the Sectoral Decarbonization Approach of Krabbe et al., which first set out how low-carbon emissions scenarios consistent with international temperature goals (e.g. well below 2C) could be disaggregated by sector such that companies' emissions pathways could be benchmarked, so to speak, and 'science-based' emissions reduction targets set.

The principal extension is to correct company emissions pathways to reflect any shortfall in past emissions reductions. That is, if a company has emitted more cumulatively than its share since the base year, it should have to reduce its future emissions by that same amount. There is more to the proposed method than this, but this is the essence of the contribution. I think this is a fair point. Without making this adjustment, carbon budgets could well be exceeded.

The paper is punchy and easy to read. However, I think the jury is still out in whether the proposed methodology is workable and under what circumstances. Below I set out various issues that I think the authors need to consider, as well as various other suggestions to improve the paper.

Major comments

Framing:

The paper is framed as a critique of ACT and TPI. This is reasonable to some extent, because ACT and TPI both constitute high-profile applications of the SDA. But ACT and TPI do much more than just apply the SDA – they provide investors and users of their data with a broad range of indicators of corporate climate action. This is especially true of ACT's complex framework, but it is also true of TPI, which now assesses companies against 19 governance/management indicators. So, I think the paper risks misrepresenting ACT and TPI. The abstract states that "these initiatives [ACT and TPI] have several limitations", but the analysis in this paper does not support a sweeping statement about these initiatives as a whole, given everything else they do. In any case, I think this paper would be better characterised as an extension of the SDA. As I explain below, I also think it is closer to SBTi in how it might work.

Data availability and scope of this method:

The proposed method requires quite a lot of detailed data about company emissions and it requires data about current and future market shares. These data are likely to be held by some companies, particularly those forward-looking companies with a clear climate strategy. Some of these data will be publicly disclosed, but some probably won't be. As such, I venture that the proposed method would fit well with an approach like SBTi, which works via company collaboration. That is, companies looking to set a 'science-based' target that is accredited by SBTi apply the method themselves using their internal data and then turn to SBTi for accreditation. The wider world doesn't see the inner workings, just the outcome (an accredited target). I think it's important to contrast this with independent data of the sort that TPI provides. TPI does not work like SBTi, although it also uses the SDA. TPI assesses companies independently based on the data they publicly disclose. Most companies do not disclose the sorts of data required to implement the method proposed in this paper. There is rarely any visibility of company emissions at the facility/asset level. How do you make projections of future market share in a way that is consistent? Companies have strategic incentives to distort their market share projection under the SDA and more broadly. That is why TPI does not deal with changing market shares. This paper uses 10 Australian electricity utilities as a proof of concept, but developed-world electricity utilities do not reveal these problems, as electricity is the easiest of all sectors on which to apply the SDA. There is a lot of data available, because in countries like Australia, as I understand it, there is a good deal of mandatory reporting of energy consumption, emissions and so on. This is just not the

case in almost any other sector. It is not the case in oil and gas, mining or any manufacturing sector I am aware of. The emissions data are often missing, or disclosed only on an intensity basis, or material emissions are missing, particularly in sectors like oil and gas where the larger part of the lifecycle carbon footprint is in Scope 3. I encourage the authors to reflect on these issues and what scope there is for application of this method. I would argue that the shortcuts taken by initiatives like TPI are a price to pay for trying to provide a large data set.

Updating:

Emissions scenarios get regularly updated. How would the proposed method deal with this? I think it could get complicated, given that the centrepiece of the method is this correction for excess historical emissions. When emissions scenarios get updated, two things happen. First, there is a host of changes to things like technology costs, demand etc., which reflect an updated and improved understanding of where the world is heading. You absolutely want to incorporate those changes. Second, account is taken of emissions in recent years. Assuming a fixed carbon budget (n.b. this can be a moving target, given evolving scientific understanding), what then happens is that past emissions are higher, and future emissions have to be lower. So excess historical emissions get embedded in the scenarios themselves. Where does that leave the companies?

How data are used:

A philosophical debate that runs through this paper is how much data to provide to investors, stakeholders etc., how much flexibility data-users are given to interpret the data, and how much we as researchers try to do the job for those users. I would argue that TPI does not really prevent users from doing similar calculations to those set out in this paper. TPI overlays company emissions pathways on low-carbon scenarios and investors can analyse those and interpret them in different ways. This is what we see in reality – a diversity of interpretations. You can take a company whose emissions intensity has been falling more slowly than a below 2C scenario in the past and you can require it to reduce its emissions faster in the future. That is doable using TPI data as set out in Figure 1, it just requires the user to do some calculations. I suspect the authors of this paper would argue that it is too much to ask your typical data-user to do this. They need help and therefore one should use the proposed method instead. I can accept this. Reasonable minds can disagree on it. But you do lose flexibility. I would encourage the authors to reflect on this issue of where the boundary lies between research production and use, and also whether the metrics they have developed would find wide application and under what circumstances. They are somewhat complex, too complex for a lot of investors I would argue.

Companies' historical responsibility:

Analogous to the issue of national responsibility for historical emissions, to what extent do the authors think that companies will accept that historical excess emissions should factor into their future emissions targets, given how few companies have set Paris-aligned targets even using the more accommodating methods of ACT/TPI? Although the base year of the IEA ETP scenarios is 2014, they were first published in 2017. So, for the first three years of the assessment period companies would have had no idea what they were going to be compared with.

Minimum action case:

What is the basis for the assumption in this case that the generating mix is 50% gas and 50% zero-carbon. Why not 60:40? Or 30:70? How would you construct a minimum action case in other sectors?

Minor comments

It is a little confusing that the method is presented in terms of absolute emissions yet Figure 1 is presented in terms of emissions intensity.

Line 47: since the Paris Agreement aims to limit global warming to well below 2C (with 1.5C as a stretch target), and well below 2C does not require reaching net zero by 2050 (this is a rule of thumb for 1.5C), it isn't strictly accurate to say the Paris Agreement requires net zero by 2050.

Line 74: although IEA scenarios happened to be used in the original implementation of the SDA, other scenarios could be used.

Lines 139-141: per my comment above, I question which elements of a firm-specific carbon intensity could be calculated by an independent research programme using only public disclosures.

Lines 211-213: this is incorrect – TPI provides full data on companies' emissions pathways set against sectoral low-carbon scenarios on its online tool. The data were aggregated into a categorical indicator in our Nature Climate Change paper.

It would help the exposition of the method to provide more intuition for what the formulae are doing, e.g. the PCP_e and PCP_i.

Simon Dietz
London
2nd Feb 2021

Reviewer #3 (Remarks to the Author):

Main feedback:

The paper proposes a methodology to benchmark and measure "transition performance" using a "strict science-based approach" and compares it to the benchmarks used in two initiatives that "have proposed methodologies to benchmark companies' performances against science-based emission reduction levels".

In doing so the paper fails to acknowledge and provide context on two important factors: 1) first that these initiatives propose to measure and assess companies alignment to Paris goals on a multi-dimensional perspective; 2) that the premises on which it basis its "strict science-approach" might be "socially flawed", due to the inherent difficulties of allocating carbon budgets on a top-down manner.

On the first point, the authors fail to acknowledge that, for example, the ACT initiative measures Paris-alignment based on a variety of quantitative and qualitative indicators. For the Electric Utilities sectors it proposes 17 quantitative indicators (only 3 of which respect to target setting, being the rest about Material investments, intangible investments, management, policy engagement and business model) and 5 qualitative indicators (Business model and strategy, Consistency and credibility, Reputation, Risk and Trend). The TPI also considers several other indicators related to management and management quality. Thus, the article does not acknowledge or discuss the multi-dimensionality of the initiatives it uses to compare its "Paris-aligned benchmark", rather narrowly focuses on a "one-dimensional" approach to measuring Paris-alignment based on a company science-based targets and then making inferences from there which are not always valid (see further below).

On what target indicators respect, the main assumption that distinguishes the approach taken in this paper is the decision to ensure that "each company strictly adheres to the Paris carbon budget". This decision, which was explicitly avoided in the construction of ACT methodologies and science-based targets, is a momentous decision whose implications are not properly discussed in the paper, namely; 1) the basis and consequences of choosing any given base year for the budget allocation exercise (paper chosen 2014), as well as the implications of choosing any given scenario or science-based target setting method, particularly at the light of what is discussed in Faria and Labutong (2019) ; 2) The influence of the market share parameter in calculating any emissions budget forward and on what basis are the future projection of activity calculated and legitimized; 3) How it would consider/not consider historical contributions and equity concerns within that allocation, a topic amply discussed in the scientific literature ; 4) how it would address carbon budget allocation for new companies and new assets; 5) carbon budget re-allocation challenges when companies have significant changes within their corporate structures, e.g. mergers and acquisitions or sales of assets, namely: would that trigger a recalculation on a re-baselined 2014 intensity or from the year of the significant change; how that re-calculation would be done; would the emission of the assets be carried to the new company carbon budget, or the old one; 6) the significant uncertainties around carbon budgets and the continuing evolution of scientific understanding of these and how, changes in these could lead to changes in the proposed company carbon budgets under the proposed method.

Finally, there is no acknowledgement of the difficulty in doing this type of top-down allocation (beyond some of the issues flagged in 1 to 6 above) and how fundamentally contradictory it is in face of the spirit of “bottom-up” governance and top-down accountability proposed by the Paris agreement . A final note on this point to recognize that the ACT Framework and ACT methodologies are a multi-stakeholder process with strong representation of companies and where this approach would be very difficult to agree. The ACT initiative was launched at COP21 with the objective of creating higher accountability and transparency on the multitude of company pledges to contribute towards Paris goals.

Addressing some issues of detail, the main criticism of ACT target indicators seems to be that they do not consider “past performance” but instead, at each assessment “recalculate” the company benchmark (CBg). Basically, the authors compare some ACT indicators on a basis of fixed budget allocation per company - when they are not supposed to operate in this way and doing it leads to distortions and invalid inferences and conclusions. The ACT indicators calculate, at each point in time an assessment is done (reporting year), a pathway for the company which is used as reference to measure some of its indicators - instead of comparing it with a previous and largely arbitrary fixed point in time.

The analysis done by the authors, assumes the scenarios and allocation mechanisms for companies will stay the same for companies, as time passes. Under this assumption new recalculated targets pathways are necessarily always more lenient and do not meet global carbon budgets – that is correct. However, ACT assessments do not assume fixed scenarios and fixed global carbon budgets in time. In fact, as carbon budgets get “consumed” and scenarios are revised, one’s expectation would be that these pathways will, necessarily, get more ambitious reflecting the a smaller volume of the carbon budget, but also improved scientific understanding of carbon budgets. Adjustments for companies’ “under-achievements” – and overall economy under-achievement - should be reflected as the climate and social scientists reflect about the implication of those facts and how to reflect them in terms of future scenarios – and not on a basis of a carbon budget set arbitrarily at some point in time. One of the characteristics of the ACT Framework and its methodologies, is that it explicitly acknowledges that it can be “plugged-in” to a variety of scenarios: “From a more general point-of-view, sector benchmarks may be changed from the 2DS scenario to another relevant low-carbon scenario depending on the availability of geography-specific, context-specific, future relevant scenarios, or future updates of the 2DS scenario – and possibly to explore higher ambition requirements. In this case, the ACT assessment report shall disclose which low-carbon background scenario has been used. This also adds some flexibility to the assessment of specific companies and to the use of the most up-to-date or more ambitious (e.g. IEA ETP B2DS) scenarios.” (page 17, ACT Framework). ACT EU benchmarks were first utilized in 2016, when only the 2DS was available. The current version of the methodology (1.1, published in early 2019) is basically an edited version for a common ACT visual layout of the 2016 methodology. This does not reflect the fact that in more recent assessments, like the ones done for the World Benchmarking Alliance, the IEA ETP B2DS has been used as reference scenario (see <https://www.worldbenchmarkingalliance.org/publication/electric-utilities/methodology/>) . Under a perspective of a fixed-based year and set carbon budget, it is true that target recalculations under same scenario and for companies that have complied with Paris-compliant trajectories would lead to over-allocation. However, if you do not follow that constraint – as ACT does not - the expectation is that, the global carbon budget constraint can still be met as scenarios get updated, reflecting both changes in the understanding of global carbon budgets (and their significant uncertainties) as well as the “consumption” of the carbon budget by economic actors. Companies’ new benchmark trajectories will be more ambitious than previous trajectories – achieving the same effect as the PCP is proposing to do, but in a different way.

Detail feedback:

The authors’ state “The ACT also strays from the most recent SDA approach in using the IEA 2DS pathway instead of the IEA B2DS pathway. Even if the ACT were to evaluate all companies’ performance since 2014 using IEA B2DS (as illustrated for the company in Figure 1b), their “previous” benchmark does not adjust for discrepancies between the actual and projected market share.” We do not totally follow what is meant by this sentence. ACT does not make a “previous” benchmark. It has specific indicators to assess if companies’ past targets have been reached or not (EU 1.3 and EU2.1 and to a certain extent EU 2.2), which fit its different philosophy on how to

assess Paris-alignment of companies.

The supplementary material is more specific and it flags as limitations "It is unspecified what year "initial" refers to, and even if it was used at a common base year consistent with the IEA pathway, it is unclear if and how the benchmark is adjusted to account for performance to date, especially how the benchmark needs to be adjusted to make up for any deficits. The ACT Framework uses "previous" and "current" benchmarks which at least provide clarity on what year the initial intensity it is based on, but here it is unclear." "Initial" refers to the "reporting year" this is, the latest year for which data is available to assess the specific indicator. The method is not explicit, but as the authors correctly inferred, there is no "adjustments for performance to date" at the level of this specific indicator – there is a specific indicator for that purpose, as mentioned above.

While the author's characterize how the proposed indicators are superior to existent ones there is no critique of the proposed indicators. Should we conclude that this is the perfect set of indicators to measure company alignment with Paris goals according to authors or are there any limitations to its use also?

ACT context

Considering that ACT initiative is taken as an important point of comparison for the article, it is important that it is properly characterized. As such, the article should address the following points: "ACT is a voluntary initiative of the UNFCCC secretariat Global Climate Agenda supporting corporate climate accountability. It develops sectoral methodologies through a multi-stakeholder process, as an accountability framework to support companies with delivering low carbon transition strategies and actions aligned with the Paris Agreement mitigation goal. ACT was launched in 2015 at COP21 (Paris) and released its ACT Framework, first 3 methodologies and benchmarks in 2016 at COP22 (Marrakesh)."

1) We should note that until recently cement companies were still using 1990 as a reference year for their inventories and emission targets. The basis for any given year – 1990, 2014, 2015, 2020 – are arbitrary and not always suitably justified.

2) Pedro Cabral Santiago Faria, Nicole Labutong, (2019) "A description of four science-based corporate GHG target-setting methods", Sustainability Accounting, Management and Policy Journal, <https://doi.org/10.1108/SAMPJ-03-2017-0031>

3) For example in country context: <https://link.springer.com/article/10.1007%2Fs10668-019-00433-1>, <https://link.springer.com/article/10.1007/s10584-016-1633-1>;

Reviewer comments, second round

Reviewer #1 (Remarks to the Author):

This paper presents new concepts and metrics related to measuring companies' CO2 emission performance against "Paris compliant" benchmarks. The authors do a relatively good job at describing the weaknesses of existing approaches (TPI and ACT) and present their own approach in a clear way using a case study. The paper makes a significant contribution to the emerging literature that address the question "what does a Paris-compliant company look like?". The paper is certainly relevant to several academic fields and corporate stakeholders (regulators, investors, etc.).

The only significant weakness of the paper, from my perspective, is the presentation of the SDA method and its use by companies in the introduction section, where a number of inaccurate claims are made (see my detailed comments below). I also think the clarity of the text presenting the limitations of SDA, TPI and ACT (around page 4-7) can be improved (see my detailed comments below). In addition, I have a list of general and specific comments that mostly relate to increasing the readability of the paper:

General:

- Your text contains a large number of abbreviations, especially in the methods section. Consider getting rid of some of the abbreviations used less frequently, to increase readability.
- Consider using more informative names than "Metric 1/2/3".
- Why is the electricity provider in Fig. 1 and 2 anonymous, given that the ten providers in Fig. 3 are disclosed? Why not be transparent about all case companies used throughout the study?

Specific:

- Line 39: More prescriptive than what? Please clarify.
- Line 70-71: I am not sure I understand the reference to the footprint concept here. Consider rewording.
- Line 72: You may also want to refer to the Context-based carbon metric (for businesses) from CSO: <https://www.sustainableorganizations.org/context-based-metrics-public-domain/>. The spreadsheet tool actually contains a function that allows users to adjust emission pathways (in order to conserve the originally developed cumulative emission target) along the lines that you suggest in your paper.
- Line 73: Can you provide a reference in support of your statement that "the Sectoral Decarbonisation Approach (SDA) is the most widely adopted"? From what I know, only a minority of companies with an SBT approved by the SBTi disclose the method that they used to set the target. In addition, I suspect that most companies with recently approved SBTs have used the ACA method, since it is applicable with 1.5 degree emission pathways and to all sectors, whereas the SDA method is currently only applicable with less ambitious emission pathways ("2 degrees" and "well-below 2 degrees") and to a handful of "homogenous" sectors.
- Line 78: For reasons given in my comment above, the statement "it can be applied to any sector" is not true. The main difference between SDA and other SBT methods is that it allows calculation of sector-specific targets, based on the sector-specific emission pathways of the IEA. But since the IEA emission pathways do not cover all sectors of the economy (agriculture is left out, for example), the SDA method cannot calculate targets for all sectors. By comparison, all other SBT methods are applicable to all sectors.
- Line 79: I am not sure that I have seen "market share" and "initial carbon intensities" recognized as "equity principles" in the literature. Can you provide a reference? You may also consider that the SDA method relies on the principles Grandfathering (the higher the initial absolute emissions, the higher the future allowed absolute emissions), Convergence (the carbon intensity of all companies are assumed to converge in 2050) and Cost-optimization (the IEA sectoral pathways have been derived by considering differences in the costs of reducing emissions across sectors).
- Line 99-115: It is unclear whether the three "key conditions" are presented by the SDA developers/SBTi or by you. For example, the SBTi does not, to my knowledge, require all companies to use 2014 as baseline year (as you note below). Please clarify the origin of the three

“key conditions”.

- Line 132: Do you mean “the updated SBTi guidance”?
- Line 135: “even if a company sets a target in 2014, there is no clarity on how to accurately include any past deficits” This is true. But SBTs are always derived from performance (emissions and activity) in a single baseline year and, as such, are independent of performance prior to that year. Given this scope, it is not surprising that SBTi does not provide clarity around “past deficits” What does “past deficits” even mean in this context? Consider rewording.
- Line 144: Revised in response to what? Emissions not following the originally targeted emission trajectory? Please clarify. Also, from Table 1 it appears that TRI does include revised pathways. Please ensure consistency between text and table.
- Line 152: Consider elaborating that this means that the ACT approach is only consistent with the least conservative interpretation of the Paris Agreement (preventing global warming of more than 2 degrees).
- Line 157: “a company’s original allocated carbon budget would be exceeded” - I think that this is the central limitation of ACT (and TPI). Perhaps you could make this clearer from Table 1. I think the point relates to your benchmark about “action deficit”, but if you were to adopt wording related to “carbon budget exceedance” (or something along those lines) you might more effectively convey your message.
- Line 203: I am confused about the last part of the sentence – “that produces a greater than zero emission intensity”. Would there be any cases where the projected production capacity in the EYF would lead to negative emission intensities? Consider rewording.
- Line 206: “the IEA B2DS allows for temporary overshoot” – overshoot of what? The underlying temperature goal? Please specify.
- Line 207: Should be “as shown in Figure 1c”, right?
- Line 212: Are you sure there should be a “hence”? I do not see the link between requiring only publicly available data and using a simple binary carbon performance metric. For example, why does the use of publicly available data prevent a more quantitative assessment that goes beyond aligned/not aligned labels?
- Line 228: “metric C” should be “metric 2C”, right?
- Fig 2.: It is a bit confusing that the “ACT – Current benchmark” has a different colour than in Fig. 1 (in which a similar red/brown colour is used for “ACT – Previous benchmark”). Please align, if possible.
- Line 249: Please remind the reader what year “immediate” refers to.
- Line 249-253: These results are shown graphically in Figure 2b, right? If so, consider referring to the figure here.
- Line 265: “Overall results are very similar to if companies followed a “maximum action case””. Starting in 2014? Please specify. The point also applies to the next sentence.
- Line 286: “The additional metrics” - Are you referring to Metrics 1, 2 and 3? Please clarify.
- Line 452: I see from the equation that you assume convergence of company emission intensity pathways in 2060 (for a given sector). This is consistent with what the SBTi used to assume. However, since their latest version of the SDA tool, incorporated in the broader SBTi tool from June 2020 (version 1.2 - <https://sciencebasedtargets.org/resources/files/SBT-Tool-v1.2.1.xlsx>), convergence is assumed to happen in 2050 (as originally assumed in Krabbe et al.). Therefore, you might want to either update your equations and results (which would not change much, I expect) or, at least, make a note about the year of convergence currently assumed by the SBTi.
- Line 481: If I understand correctly, you propose to use “regional sectoral activity projections” and assume a constant market share for each company within? This raises two questions: 1) What is the data source for these “regional sectoral activity projections”? - I do not see this mentioned in your Data section. 2) Why not use company-specific projections, if available? – Such projections may either be made by companies themselves or external analysts and are usually available in financial databases, such as Bloomberg and FactSet, though usually only extending around 3 years into the future (which is a big limitation in this context, I know). I encourage you to address these questions in the Methods text.
- Line 542: I am not sure that I understand what “The metric” refers to. The previous sentence is about energy generation capacity, but this sentence is about emission trajectories, right? Consider rewording.
- Line 600: Does this also mean that the SDA method is not applicable to hydro power companies? Consider making a note about this.

Reviewer #2 (Remarks to the Author):

This paper contributes to an emerging literature on assessing corporate climate action. It is an important topic, and more academic scholarship and constructive debate is urgently needed, particularly since a lot of research is published in the grey literature rather than in peer-reviewed academic journals.

This paper builds on the Sectoral Decarbonization Approach of Krabbe et al., which first set out how low-carbon emissions scenarios consistent with international temperature goals (e.g. well below 2C) could be disaggregated by sector such that companies' emissions pathways could be benchmarked, so to speak, and 'science-based' emissions reduction targets set.

The principal extension is to correct company emissions pathways to reflect any shortfall in past emissions reductions. That is, if a company has emitted more cumulatively than its share since the base year, it should have to reduce its future emissions by that same amount. There is more to the proposed method than this, but this is the essence of the contribution. I think this is a fair point. Without making this adjustment, carbon budgets could well be exceeded.

The paper is punchy and easy to read. However, I think the jury is still out in whether the proposed methodology is workable and under what circumstances. Below I set out various issues that I think the authors need to consider, as well as various other suggestions to improve the paper.

Major comments

Framing:

The paper is framed as a critique of ACT and TPI. This is reasonable to some extent, because ACT and TPI both constitute high-profile applications of the SDA. But ACT and TPI do much more than just apply the SDA – they provide investors and users of their data with a broad range of indicators of corporate climate action. This is especially true of ACT's complex framework, but it is also true of TPI, which now assesses companies against 19 governance/management indicators. So, I think the paper risks misrepresenting ACT and TPI. The abstract states that "these initiatives [ACT and TPI] have several limitations", but the analysis in this paper does not support a sweeping statement about these initiatives as a whole, given everything else they do. In any case, I think this paper would be better characterised as an extension of the SDA. As I explain below, I also think it is closer to SBTi in how it might work.

Data availability and scope of this method:

The proposed method requires quite a lot of detailed data about company emissions and it requires data about current and future market shares. These data are likely to be held by some companies, particularly those forward-looking companies with a clear climate strategy. Some of these data will be publicly disclosed, but some probably won't be. As such, I venture that the proposed method would fit well with an approach like SBTi, which works via company collaboration. That is, companies looking to set a 'science-based' target that is accredited by SBTi apply the method themselves using their internal data and then turn to SBTi for accreditation. The wider world doesn't see the inner workings, just the outcome (an accredited target). I think it's important to contrast this with independent data of the sort that TPI provides. TPI does not work like SBTi, although it also uses the SDA. TPI assesses companies independently based on the data they publicly disclose. Most companies do not disclose the sorts of data required to implement the method proposed in this paper. There is rarely any visibility of company emissions at the facility/asset level. How do you make projections of future market share in a way that is consistent? Companies have strategic incentives to distort their market share projection under the SDA and more broadly. That is why TPI does not deal with changing market shares. This paper uses 10 Australian electricity utilities as a proof of concept, but developed-world electricity utilities do not reveal these problems, as electricity is the easiest of all sectors on which to apply the SDA. There is a lot of data available, because in countries like Australia, as I understand it, there is a good deal of mandatory reporting of energy consumption, emissions and so on. This is just not the

case in almost any other sector. It is not the case in oil and gas, mining or any manufacturing sector I am aware of. The emissions data are often missing, or disclosed only on an intensity basis, or material emissions are missing, particularly in sectors like oil and gas where the larger part of the lifecycle carbon footprint is in Scope 3. I encourage the authors to reflect on these issues and what scope there is for application of this method. I would argue that the shortcuts taken by initiatives like TPI are a price to pay for trying to provide a large data set.

Updating:

Emissions scenarios get regularly updated. How would the proposed method deal with this? I think it could get complicated, given that the centrepiece of the method is this correction for excess historical emissions. When emissions scenarios get updated, two things happen. First, there is a host of changes to things like technology costs, demand etc., which reflect an updated and improved understanding of where the world is heading. You absolutely want to incorporate those changes. Second, account is taken of emissions in recent years. Assuming a fixed carbon budget (n.b. this can be a moving target, given evolving scientific understanding), what then happens is that past emissions are higher, and future emissions have to be lower. So excess historical emissions get embedded in the scenarios themselves. Where does that leave the companies?

How data are used:

A philosophical debate that runs through this paper is how much data to provide to investors, stakeholders etc., how much flexibility data-users are given to interpret the data, and how much we as researchers try to do the job for those users. I would argue that TPI does not really prevent users from doing similar calculations to those set out in this paper. TPI overlays company emissions pathways on low-carbon scenarios and investors can analyse those and interpret them in different ways. This is what we see in reality – a diversity of interpretations. You can take a company whose emissions intensity has been falling more slowly than a below 2C scenario in the past and you can require it to reduce its emissions faster in the future. That is doable using TPI data as set out in Figure 1, it just requires the user to do some calculations. I suspect the authors of this paper would argue that it is too much to ask your typical data-user to do this. They need help and therefore one should use the proposed method instead. I can accept this. Reasonable minds can disagree on it. But you do lose flexibility. I would encourage the authors to reflect on this issue of where the boundary lies between research production and use, and also whether the metrics they have developed would find wide application and under what circumstances. They are somewhat complex, too complex for a lot of investors I would argue.

Companies' historical responsibility:

Analogous to the issue of national responsibility for historical emissions, to what extent do the authors think that companies will accept that historical excess emissions should factor into their future emissions targets, given how few companies have set Paris-aligned targets even using the more accommodating methods of ACT/TPI? Although the base year of the IEA ETP scenarios is 2014, they were first published in 2017. So, for the first three years of the assessment period companies would have had no idea what they were going to be compared with.

Minimum action case:

What is the basis for the assumption in this case that the generating mix is 50% gas and 50% zero-carbon. Why not 60:40? Or 30:70? How would you construct a minimum action case in other sectors?

Minor comments

It is a little confusing that the method is presented in terms of absolute emissions yet Figure 1 is presented in terms of emissions intensity.

Line 47: since the Paris Agreement aims to limit global warming to well below 2C (with 1.5C as a stretch target), and well below 2C does not require reaching net zero by 2050 (this is a rule of thumb for 1.5C), it isn't strictly accurate to say the Paris Agreement requires net zero by 2050.

Line 74: although IEA scenarios happened to be used in the original implementation of the SDA, other scenarios could be used.

Lines 139-141: per my comment above, I question which elements of a firm-specific carbon intensity could be calculated by an independent research programme using only public disclosures.

Lines 211-213: this is incorrect – TPI provides full data on companies’ emissions pathways set against sectoral low-carbon scenarios on its online tool. The data were aggregated into a categorical indicator in our Nature Climate Change paper.

It would help the exposition of the method to provide more intuition for what the formulae are doing, e.g. the PCP_e and PCP_i.

Simon Dietz
London
2nd Feb 2021

Reviewer #3 (Remarks to the Author):

Main feedback:

The paper proposes a methodology to benchmark and measure “transition performance” using a “strict science-based approach” and compares it to the benchmarks used in two initiatives that “have proposed methodologies to benchmark companies’ performances against science-based emission reduction levels”.

In doing so the paper fails to acknowledge and provide context on two important factors: 1) first that these initiatives propose to measure and assess companies alignment to Paris goals on a multi-dimensional perspective; 2) that the premises on which it basis its “strict science-approach” might be “socially flawed”, due to the inherent difficulties of allocating carbon budgets on a top-down manner.

On the first point, the authors fail to acknowledge that, for example, the ACT initiative measures Paris-alignment based on a variety of quantitative and qualitative indicators. For the Electric Utilities sectors it proposes 17 quantitative indicators (only 3 of which respect to target setting, being the rest about Material investments, intangible investments, management, policy engagement and business model) and 5 qualitative indicators (Business model and strategy, Consistency and credibility, Reputation, Risk and Trend). The TPI also considers several other indicators related to management and management quality. Thus, the article does not acknowledge or discuss the multi-dimensionality of the initiatives it uses to compare its “Paris-aligned benchmark”, rather narrowly focuses on a “one-dimensional” approach to measuring Paris-alignment based on a company science-based targets and then making inferences from there which are not always valid (see further below).

On what target indicators respect, the main assumption that distinguishes the approach taken in this paper is the decision to ensure that “each company strictly adheres to the Paris carbon budget”. This decision, which was explicitly avoided in the construction of ACT methodologies and science-based targets, is a momentous decision whose implications are not properly discussed in the paper, namely; 1) the basis and consequences of choosing any given base year for the budget allocation exercise (paper chosen 2014), as well as the implications of choosing any given scenario or science-based target setting method, particularly at the light of what is discussed in Faria and Labutong (2019) ; 2) The influence of the market share parameter in calculating any emissions budget forward and on what basis are the future projection of activity calculated and legitimized; 3) How it would consider/not consider historical contributions and equity concerns within that allocation, a topic amply discussed in the scientific literature ; 4) how it would address carbon budget allocation for new companies and new assets; 5) carbon budget re-allocation challenges when companies have significant changes within their corporate structures, e.g. mergers and acquisitions or sales of assets, namely: would that trigger a recalculation on a re-baselined 2014 intensity or from the year of the significant change; how that re-calculation would be done; would the emission of the assets be carried to the new company carbon budget, or the old one; 6) the significant uncertainties around carbon budgets and the continuing evolution of scientific understanding of these and how, changes in these could lead to changes in the proposed company carbon budgets under the proposed method.

Finally, there is no acknowledgement of the difficulty in doing this type of top-down allocation (beyond some of the issues flagged in 1 to 6 above) and how fundamentally contradictory it is in face of the spirit of “bottom-up” governance and top-down accountability proposed by the Paris agreement . A final note on this point to recognize that the ACT Framework and ACT methodologies are a multi-stakeholder process with strong representation of companies and where this approach would be very difficult to agree. The ACT initiative was launched at COP21 with the objective of creating higher accountability and transparency on the multitude of company pledges to contribute towards Paris goals.

Addressing some issues of detail, the main criticism of ACT target indicators seems to be that they do not consider “past performance” but instead, at each assessment “recalculate” the company benchmark (CBg). Basically, the authors compare some ACT indicators on a basis of fixed budget allocation per company - when they are not supposed to operate in this way and doing it leads to distortions and invalid inferences and conclusions. The ACT indicators calculate, at each point in time an assessment is done (reporting year), a pathway for the company which is used as reference to measure some of its indicators - instead of comparing it with a previous and largely arbitrary fixed point in time.

The analysis done by the authors, assumes the scenarios and allocation mechanisms for companies will stay the same for companies, as time passes. Under this assumption new recalculated targets pathways are necessarily always more lenient and do not meet global carbon budgets – that is correct. However, ACT assessments do not assume fixed scenarios and fixed global carbon budgets in time. In fact, as carbon budgets get “consumed” and scenarios are revised, one’s expectation would be that these pathways will, necessarily, get more ambitious reflecting the a smaller volume of the carbon budget, but also improved scientific understanding of carbon budgets. Adjustments for companies’ “under-achievements” – and overall economy under-achievement - should be reflected as the climate and social scientists reflect about the implication of those facts and how to reflect them in terms of future scenarios – and not on a basis of a carbon budget set arbitrarily at some point in time. One of the characteristics of the ACT Framework and its methodologies, is that it explicitly acknowledges that it can be “plugged-in” to a variety of scenarios: “From a more general point-of-view, sector benchmarks may be changed from the 2DS scenario to another relevant low-carbon scenario depending on the availability of geography-specific, context-specific, future relevant scenarios, or future updates of the 2DS scenario – and possibly to explore higher ambition requirements. In this case, the ACT assessment report shall disclose which low-carbon background scenario has been used. This also adds some flexibility to the assessment of specific companies and to the use of the most up-to-date or more ambitious (e.g. IEA ETP B2DS) scenarios.” (page 17, ACT Framework). ACT EU benchmarks were first utilized in 2016, when only the 2DS was available. The current version of the methodology (1.1, published in early 2019) is basically an edited version for a common ACT visual layout of the 2016 methodology. This does not reflect the fact that in more recent assessments, like the ones done for the World Benchmarking Alliance, the IEA ETP B2DS has been used as reference scenario (see <https://www.worldbenchmarkingalliance.org/publication/electric-utilities/methodology/>) . Under a perspective of a fixed-based year and set carbon budget, it is true that target recalculations under same scenario and for companies that have complied with Paris-compliant trajectories would lead to over-allocation. However, if you do not follow that constraint – as ACT does not - the expectation is that, the global carbon budget constraint can still be met as scenarios get updated, reflecting both changes in the understanding of global carbon budgets (and their significant uncertainties) as well as the “consumption” of the carbon budget by economic actors. Companies’ new benchmark trajectories will be more ambitious than previous trajectories – achieving the same effect as the PCP is proposing to do, but in a different way.

Detail feedback:

The authors’ state “The ACT also strays from the most recent SDA approach in using the IEA 2DS pathway instead of the IEA B2DS pathway. Even if the ACT were to evaluate all companies’ performance since 2014 using IEA B2DS (as illustrated for the company in Figure 1b), their “previous” benchmark does not adjust for discrepancies between the actual and projected market share.” We do not totally follow what is meant by this sentence. ACT does not make a “previous” benchmark. It has specific indicators to assess if companies’ past targets have been reached or not (EU 1.3 and EU2.1 and to a certain extent EU 2.2), which fit its different philosophy on how to

assess Paris-alignment of companies.

The supplementary material is more specific and it flags as limitations "It is unspecified what year "initial" refers to, and even if it was used at a common base year consistent with the IEA pathway, it is unclear if and how the benchmark is adjusted to account for performance to date, especially how the benchmark needs to be adjusted to make up for any deficits. The ACT Framework uses "previous" and "current" benchmarks which at least provide clarity on what year the initial intensity it is based on, but here it is unclear." "Initial" refers to the "reporting year" this is, the latest year for which data is available to assess the specific indicator. The method is not explicit, but as the authors correctly inferred, there is no "adjustments for performance to date" at the level of this specific indicator – there is a specific indicator for that purpose, as mentioned above.

While the author's characterize how the proposed indicators are superior to existent ones there is no critique of the proposed indicators. Should we conclude that this is the perfect set of indicators to measure company alignment with Paris goals according to authors or are there any limitations to its use also?

ACT context

Considering that ACT initiative is taken as an important point of comparison for the article, it is important that it is properly characterized. As such, the article should address the following points: "ACT is a voluntary initiative of the UNFCCC secretariat Global Climate Agenda supporting corporate climate accountability. It develops sectoral methodologies through a multi-stakeholder process, as an accountability framework to support companies with delivering low carbon transition strategies and actions aligned with the Paris Agreement mitigation goal. ACT was launched in 2015 at COP21 (Paris) and released its ACT Framework, first 3 methodologies and benchmarks in 2016 at COP22 (Marrakesh)."

1) We should note that until recently cement companies were still using 1990 as a reference year for their inventories and emission targets. The basis for any given year – 1990, 2014, 2015, 2020 – are arbitrary and not always suitably justified.

2) Pedro Cabral Santiago Faria, Nicole Labutong, (2019) "A description of four science-based corporate GHG target-setting methods", Sustainability Accounting, Management and Policy Journal, <https://doi.org/10.1108/SAMPJ-03-2017-0031>

3) For example in country context: <https://link.springer.com/article/10.1007%2Fs10668-019-00433-1>, <https://link.springer.com/article/10.1007/s10584-016-1633-1>;

POINT BY POINT RESPONSE TO REVIEWER COMMENTS

Reviewer #1 (Remarks to the Author):	Response to Reviewer 1:
This paper presents new concepts and metrics related to measuring companies' CO2 emission performance against "Paris compliant" benchmarks. The authors do a relatively good job at describing the weaknesses of existing approaches (TPI and ACT) and present their own approach in a clear way using a case study. The paper makes a significant contribution to the emerging literature that address the question "what does a Paris-compliant company look like?". The paper is certainly relevant to several academic fields and corporate stakeholders (regulators, investors, etc.). The only significant weakness of the paper, from my perspective, is the presentation of the SDA method and its use by companies in the introduction section, where a number of inaccurate claims are made (see my detailed comments below). I also think the clarity of the text presenting the limitations of SDA, TPI and ACT (around page 4-7) can be improved (see my detailed comments below). In addition, I have a list of general and specific comments that mostly relate to increasing the readability of the paper:	We thank the reviewer for the clear summary of our paper and highlighting its wide contribution. We also thank the reviewer for identifying the main weakness of the paper, which has been instrumental in our reflections and the consequential re-direction and strengthening of the paper. The SDA is indeed no longer the main method used by companies to set science-based targets, the Absolute Contraction method is. The Science Based Targets initiative also only recommends using the SDA for certain sectors. We have considered the feedback from all the reviewers and have redirected the paper accordingly. Rather than focusing solely on the SDA method, we now focus on conditions that are required for any such allocation methodology to be used to measure "Paris Compliance". We apply this to several allocation methodologies, including the Absolute Contraction Approach and the Context-based Carbon metric that was highlighted in the reviewer's comments. We have consequently moved the comparison of the PCP to the TPI and ACT to the supplementary materials. We appreciate the reviewers' detailed comments and have made changes to improve the manuscript's accuracy. We reply to each comment in more detail below.
General:	
 •Your text contains a large number of abbreviations, especially in the methods section. Consider getting rid of some of the abbreviations used less frequently, to increase readability. 	We thank the reviewer for this comment and have removed the abbreviation Company Carbon Budget, and the International Energy Agency. Further, we clarified the PCPi to PCP_{intensity} and PCPe to PCP_{emissions}.
 •Consider using more informative names than "Metric 1/2/3". 	We agree that this would improve readability, and thus have renamed them Metric 1: "Metric 1: Performance to Date" Metric 2: "Metric 2: Projected Performance" Metric 3: "Metric 3: Re-alignment Decarbonisation Rate" see lines 249-251

 •Why is the electricity provider in Fig. 1 and 2 anonymous, given that the ten providers in Fig. 3 are disclosed? Why not be transparent about all case companies used throughout the study? 	This is a good point. We no longer refer to the company as “anonymous” and simply as “AGL, an Australian electricity provider” (line 266) and specify in Fig 1 that it is “the largest Australian electric utility company (AGL)”. (line 285-286)
Specific:	
 •Line 39: More prescriptive than what? Please clarify. 	We realise this was not clear and thank the reviewer for pointing this out. As we have changed the angle of the paper, we no longer make the comparison with TPI and ACT in the main body of the text. Therefore, this sentence was removed.
 •Line 70-71: I am not sure I understand the reference to the footprint concept here. Consider rewording. 	Initially we re-phrased this to “their sector and geographic location” to clarify this, but then it was removed as part of our re-direction of the paper.
 •Line 72: You may also want to refer to the Context-based carbon metric (for businesses) from CSO: https://www.sustainableorganizations.org/context-based-metrics-public-domain/. The spreadsheet tool actually contains a function that allows users to adjust emission pathways (in order to conserve the originally developed cumulative emission target) along the lines that you suggest in your paper. 	We thank the reviewer for informing us of the Context-based Carbon metric from CSO. Incorporating the CSO is one of the reasons we have completed a wider reflection on what it means for a company to be “Paris-Compliant”, which we believe is an instrumental step in reshaping and strengthening the paper. We have now included the Context-based Carbon Metric method developed by the Centre for Sustainable Organisations (CSO) within the paper (see Table 1).
 •Line 73: Can you provide a reference in support of your statement that “the Sectoral Decarbonisation Approach (SDA) is the most widely adopted”? From what I know, only a minority of companies with an SBT approved by the SBTi disclose the method that they used to set the target. In addition, I suspect that most companies with recently approved SBTs have used the ACA method, since it is applicable with 1.5 degree emission pathways and to all sectors, whereas the SDA method is currently only applicable with less ambitious emission pathways (“2 degrees” and “well-below 2 degrees”) and to a handful of “homogenous” sectors. 	The narrow focus on the SDA method was a key limitation identified in our previous manuscript. We agree that there has been a move towards using the ACA and we have confirmed this by assessing data on the companies that have set targets to date through the Science Based Target initiative. We have accordingly corrected our statement and now introduce the SDA as: “The SDA was the first method adopted and co-developed by the Science Based Targets initiative (an initiative by the Carbon Disclosure Project, World Wide Fund for nature, UN Global Compact, and World Resources Institute) in 2015.” (line 144-146).

•Line 78: For reasons given in my comment above, the statement “it can be applied to any sector” is not true. The main difference between SDA and other SBT methods is that it allows calculation of sector-specific targets, based on the sector-specific emission pathways of the IEA. But since the IEA emission pathways do not cover all sectors of the economy (agriculture is left out, for example), the SDA method cannot calculate targets for all sectors. By comparison, all other SBT methods are applicable to all sectors.	The SDA originally specified that companies that belong in “other industry” (one not specified by the sectoral IEA pathways), or are heterogenous, should use tCO2 per 2011 US\$, which is based on the GEVA method (Krabbe et al., 2015). We further clarify how the Science Based Targets initiative has changed this: “Initially, companies in “other industries” (for which sectoral pathways were not available) were suggested to use the Greenhouse gas Emissions per unit of Value Added (GEVA) method (Krabbe et al., 2015), which allocates the carbon budget to companies according to their economic activity. However, the Science Based Targets initiative has moved away from this approach and is instead suggesting the Absolute Contraction Approach (ACA) for these companies, applying an equal percentage of emission reductions to every company.” (line 150-156)
•Line 79: I am not sure that I have seen “market share” and “initial carbon intensities” recognized as “equity principles” in the literature. Can you provide a reference? You may also consider that the SDA method relies on the principles Grandfathering (the higher the initial absolute emissions, the higher the future allowed absolute emissions), Convergence (the carbon intensity of all companies are assumed to converge in 2050) and Cost-optimization (the IEA sectoral pathways have been derived by considering differences in the costs of reducing emissions across sectors).	The reviewer makes a good point. We have looked into equity principles and have now added this as a third, desirable but not necessary, condition around equity. We articulate that: “Finally, a desirable, but not necessary condition is that the methodology should account for “common but differentiated responsibilities” in addressing climate change (Paris Agreement, Article 2, Part 2¹), meaning that companies in developed nations have a greater responsibility in mitigating climate change than companies in developing countries. The methodology should be clear on how it allocates the budget amongst companies in different countries” (line 128-132) We have further noted the grandfathering used by the SDA and other methods, as this gives insight into how the carbon budget is allocated amongst companies (see Table 1). We also refer to it as part of the limitations of the SDA method: “It should be noted that the SDA method itself also has limitations in its foundations, such as allowing companies with a higher initial intensity to have a greater share of the remaining global carbon budget (i.e. grandfathering).” (line 361-364)

¹ “This Agreement will be implemented to reflect equity and the principle of common but differentiated responsibilities and respective capabilities, in the light of different national circumstances.” Part 2 of Article 2 of Paris Agreement

•Line 99-115: It is unclear whether the three “key conditions” are presented by the SDA developers/SBTi or by you. For example, the SBTi does not, to my knowledge, require all companies to use 2014 as baseline year (as you note below). Please clarify the origin of the three “key conditions”.	We have removed this section and replaced it with the conditions an allocation method must meet to be considered as “Paris Compliant”.
•Line 132: Do you mean “the updated SBTi guidance”?	Yes, thank you! This has been corrected.
•Line 135: “even if a company sets a target in 2014, there is no clarity on how to accurately include any past deficits” This is true. But SBTs are always derived from performance (emissions and activity) in a single baseline year and, as such, are independent of performance prior to that year. Given this scope, it is not surprising that SBTi does not provide clarity around “past deficits” What does “past deficits” even mean in this context? Consider rewording.	Thank you for pointing out our need to clarify. We were referring to deficits prior to the 2014 base year. We have now reworded this as “any deficits it incurs since this base year” (line 208-209)
•Line 144: Revised in response to what? Emissions not following the originally targeted emission trajectory? Please clarify. Also, from Table 1 it appears that TRI does include revised pathways. Please ensure consistency between text and table.	Good point. We clarify that it is revised in response to the originally targeted emissions not being met (and adjusted for any deficits). Table 1 states that the TPI does not have a revised pathway, therefore is it in line with text. We have changed the wording to clarify that it is a “re-alignment” pathway.
•Line 152: Consider elaborating that this means that the ACT approach is only consistent with the least conservative interpretation of the Paris Agreement (preventing global warming of more than 2 degrees).	Initially we added: “, thus only consistent with the least conservative interpretation of the Paris Agreement.” to the main text, but this section was subsequently moved to the supplementary materials. We further highlight this point when discussing the limitations of the SDA method, where we added “The B2DS pathway is also only consistent with the more conservative interpretation of the Paris Agreement with a 50% chance of keeping global warming below 1.75°C.” (line 368-370).
•Line 157: “a company’s original allocated carbon budget would be exceeded” - I think that this is the central limitation of ACT (and TPI). Perhaps you could make this clearer from Table 1. I think the point relates to your benchmark about “action deficit”, but if you were to adopt wording related to “carbon budget exceedance” (or something along those lines) you might more effectively convey your message.	This is now the second requirement of operationalisation. We refer to carbon budget exceedance on line 177-183: “The second operationalisation requirement is that if a company does not meet the emissions reductions of their PCP in any year, they must reconstruct a “re-alignment” PCP that adheres to all the conditions above while maintaining the company’s carbon budget. This will mean that any company which fails to reduce their emissions sufficiently in any year must increase their reductions in subsequent years to compensate. In short, if a company has not met its

	target, a “re-alignment” pathway should be defined to make up for the action deficit.” We highlight the issue within the SBTi guidance, which although now acknowledging in principle the ‘action deficit’ does not provide specific “clarity on how to accurately include any deficits it incurs since this base year, or account for changes in their actual vs projected market share, in their target recalculation.” (line 208-210).
•Line 203: I am confused about the last part of the sentence – “that produces a greater than zero emission intensity”. Would there be any cases where the projected production capacity in the EYF would lead to negative emission intensities? Consider rewording.	We meant that this statistic is only relevant if the company’s production at that point in time is still associated with a positive emission intensity – as an indication of how much production would need to be stranded to stay within their carbon budget. We have now rephrased this sentence as follows: “2b) their projected (carbon positive) production in the EYF” (line 258-259)
•Line 206: “the IEA B2DS allows for temporary overshoot” – overshoot of what? The underlying temperature goal? Please specify.	Clarified to “temporary overshoot of the carbon budget” (line 263)
•Line 207: Should be “as shown in Figure 1c”, right?	Correct, thank you for picking this up. Yet this sentence has been removed when moving the comparison to the TPI and act to the Supplementary Materials.
•Line 212: Are you sure there should be a “hence”? I do not see the link between requiring only publicly available data and using a simple binary carbon performance metric. For example, why does the use of publically available data prevent a more quantitative assessment that goes beyond aligned/not aligned labels?	We agree, “hence” has been removed from the sentence (Supplementary Materials, line 72).
•Line 228: “metric C” should be “metric 2C”, right?	Yes, thank you! This has been corrected.
•Fig 2.: It is a bit confusing that the “ACT – Current benchmark” has a different colour than in Fig. 1 (in which a similar red/brown colour is used for “ACT – Previous benchmark”). Please align, if possible.	Thank you - this has been fixed in the Supplementary Materials where most of the comparison with the ACT and TPI now takes place.
•Line 249: Please remind the reader what year “immediate” refers to.	We have added a clarification to this sentence, which now reads: “If the company takes immediate action (in this case, in 2020) to realign its trajectory to stay within its carbon budget to 2050, it will need to decarbonise its operations 1.46 times more rapidly than had it followed its PCP since 2014 (Table 2, Figure 2).” (line 303-306)
•Line 249-253: These results are shown graphically in Figure 2b, right? If so, consider referring to the figure here.	Correct. We have added in a reference to Figure 2 here: “If the company takes immediate action (in this case, in 2020) to realign its trajectory to stay

	within its carbon budget to 2050, it will need to decarbonise its operations 1.46 times more rapidly than had it followed its PCP since 2014 (Table 2, Figure 2)." (line 303-306)
•Line 265: "Overall results are very similar to if companies followed a "maximum action case"". Starting in 2014? Please specify. The point also applies to the next sentence.	This is from 2019 onwards, which is the last year we had actual data available. We have now clarified this in the sentence prior, by referring to the year 2019 twice, so that it is clear in the following sentences what we are referring to: "In 2019, all electric utility companies currently exceeded their PCP (Fig 2a), and if the companies followed a "minimum action case", from 2019 onwards, all but one company is expected to exceed their total carbon budget before 2030 (Fig 2b) and exceed their carbon budget by between 2 and 4 times in the year 2050 (Extended Data Figure 1)" (line 320-324)
•Line 286: "The additional metrics" - Are you referring to Metrics 1, 2 and 3? Please clarify.	Clarified by rephrasing the sentence to: "Our three proposed metrics can be used by companies, investors, and other stakeholders to evaluate the company's transition performance" (line 348-350)
•Line 452: I see from the equation that you assume convergence of company emission intensity pathways in 2060 (for a given sector). This is consistent with what the SBTi used to assume. However, since their latest version of the SDA tool, incorporated in the broader SBTi tool from June 2020 (version 1.2 - https://sciencebasedtargets.org/resources/files/SBT-Tool-v1.2.1.xlsx), convergence is assumed to happen in 2050 (as originally assumed in Krabbe et al.). Therefore, you might want to either update your equations and results (which would not change much, I expect) or, at least, make a note about the year of convergence currently assumed by the SBTi.	We thank the reviewer for the attention to detail, this ensures we are doing everything correctly and are making sure it is clear how it compares to the SBTi. In the SBTi tool referred to in the comment, the "Calculations" and "Database" tab both use a pathway until 2060 (consistent with IEA B2DS), but in the "SBT Tool" tab the targets go indeed until 2050 maximum. To our knowledge, there is no explanation for this in any of the SBTi documentation. We now note this difference in the methods section "This is slightly different to the approach of the SBTi, who uses 2050 as the year of convergence for all the sectors." (line 625-626)
•Line 481: If I understand correctly, you propose to use "regional sectoral activity projections" and assume a constant market share for each company within? This raises two questions: 1) What is the data source for these "regional sectoral activity projections"? - I do not see this mentioned in your Data section. 2) Why not use company-specific projections, if available? – Such projections may either be made by companies themselves or external analysts and are usually available in financial databases, such as Bloomberg and FactSet, though usually only extending around 3 years into the future	Thank you for these questions and insights. For question 1), the data source is the IEA for which we took the OECD projections. Initially, we thought the non-OECD countries had more granular regional activity available, but not for the sectors identified by the SDA. Therefore, it is regional only in a sense of OECD or non-OECD. We have therefore rephrased the sentence as follows:

(which is a big limitation in this context, I know). I encourage you to address these questions in the Methods text.

“For the actual PCP, m_y and CA_y is known, for the projected PCP, m_y and CA_y is projected, keeping market share constant according to sectoral activity projections (OECD or non-OECD).” (line 577-578)

For question 2), we agree that using actual projections would be better. However, production itself, and thus also production projections are actually not so easily available – which is a significant limitation of using the SDA method. We highlight this on line 370-377: *“Further, company data availability is a concern in calculating PCP’s, especially for the SDA method as it requires production output in addition to emissions. We deliberately demonstrated PCPs with a data rich set of companies (Australian energy providers) and a data poor set (global cement companies) to demonstrate PCPs could still be adequately applied. Furthermore, the companies themselves can still make the necessary calculations to provide stakeholders interested in their Paris compliance, and our PCP approach provides directions to stakeholders on what data to request from companies.”*

We also discuss the issue of data availability surrounding production levels in the methods section. Specifically, we note in line 605-607: *“We understand that these trajectories cannot easily be used for other sectors, where plant/asset data and intensity are not available. Also, ideally real company projections are used to estimate these trajectories. If this is not available, we suggest projecting forward using other estimation methods proposed in Rekker et al. (2018), such as recent emission growth or reduction rates. For the cement companies in Figure 2 (Fig 2d, e, f) we have projected forward using the 2014-2019 Carbon Intensity geometric growth/reduction rate.”*

•Line 542: I am not sure that I understand what “The metric” refers to. The previous sentence is about energy generation capacity, but this sentence is about emission trajectories, right? Consider rewording.

Thank you for picking up this important distinction. The metric in fact only captures the production levels at that point, not the production capacity. We have changed this throughout the text.

	Specifically: “Metric 2b. This metric measures the production levels (CAy) that are estimated to be still in place when the company reaches its EYF (metric 2a). This is a measure of how much energy generation capacity will need to be retired at the EYF if the company is to continue with its projected outputs but stay within its carbon budget. Metric 2b is observed from the difference in the reference emission trajectory in the EYF and the reference emission trajectory the year prior.” (line 640-644)
•Line 600: Does this also mean that the SDA method is not applicable to hydro power companies? Consider making a note about this.	The decision to choose 50/50 was arbitrary. We have removed this case.

Reviewer #2 (Remarks to the Author):	Response to Reviewer 2:
This paper contributes to an emerging literature on assessing corporate climate action. It is an important topic, and more academic scholarship and constructive debate is urgently needed, particularly since a lot of research is published in the grey literature rather than in peer-reviewed academic journals. This paper builds on the Sectoral Decarbonization Approach of Krabbe et al., which first set out how low-carbon emissions scenarios consistent with international temperature goals (e.g. well below 2C) could be disaggregated by sector such that companies' emissions pathways could be benchmarked, so to speak, and 'science-based' emissions reduction targets set. The principal extension is to correct company emissions pathways to reflect any shortfall in past emissions reductions. That is, if a company has emitted more cumulatively than its share since the base year, it should have to reduce its future emissions by that same amount. There is more to the proposed method than this, but this is the essence of the contribution. I think this is a fair point. Without making this adjustment, carbon budgets could well be exceeded. The paper is punchy and easy to read. However, I think the jury is still out in whether the proposed methodology is workable and under what circumstances. Below I set out various issues that I think the authors need to consider, as well as various other suggestions to improve the paper.	We thank the reviewer for acknowledging the importance of academic debate on this topic. This has helped us be much clearer about the exact aim of our paper – to specify what is required and to provide a clear framework for assessing whether companies are aligned with the Paris goals through academic examination. We also thank the reviewer for summarizing the core of the paper “if a company has emitted more cumulatively than its share since the base year, it should have to reduce its future emissions by that same amount.” and “Without making this adjustment, carbon budgets could well be exceeded.”. We have considered the feedback from all the reviewers and have redirected the paper accordingly. Rather than focusing solely on the SDA method, we now focus on conditions that are required for any such allocation methodology to be used to measure “Paris Compliance” in terms of the Paris goals. We apply this to several allocation methodologies. We then set our requirements for operationalising the PCPs, which includes the core element of companies needing to compensate for previous shortfalls. We have moved the comparison of the PCP to the TPI and ACT to the supplementary materials accordingly, as this was not intended to form the core message of our manuscript. We thank the reviewer for his suggestions which have greatly helped in improving the manuscript. We also really appreciate the structure of the review with the categories of major and minor comments, and headings for the items. This allows us to respond to the comments in a structured way. Please find a detailed response to each item below.
Major comments Framing: The paper is framed as a critique of ACT and TPI. This is reasonable to some extent, because ACT and TPI both constitute high-profile applications of the SDA. But ACT and TPI do much more than just apply the SDA –	We agree that it was not clear from our paper that the TPI and ACT include a wide range of metrics to examine corporate action on climate change. We have addressed this by clearly setting the scope of our paper, which focuses on quantitative metrics, and by acknowledging that qualitative metrics are also useful, particularly for evaluating the likelihood of quantitative goals

they provide investors and users of their data with a broad range of indicators of corporate climate action. This is especially true of ACT’s complex framework, but it is also true of TPI, which now assesses companies against 19 governance/management indicators. So, I think the paper risks misrepresenting ACT and TPI. The abstract states that “these initiatives [ACT and TPI] have several limitations”, but the analysis in this paper does not support a sweeping statement about these initiatives as a whole, given everything else they do. In any case, I think this paper would be better characterised as an extension of the SDA. As I explain below, I also think it is closer to SBTi in how it might work.	being met. From a climate perspective, only the actual emissions matter, and hence we focus on how emissions for our analysis. We have clarified this in the text as follows “There are two recent initiatives that assess a company’s transition performance using the SDA method: 1) The Transition Pathway Initiative (TPI) [25]; and 2) Assessing low-Carbon Transitions (ACT) [26]. These two initiatives include a wide variety of qualitative and quantitative metrics to assess a companies’ performance. Acknowledging the importance of the variety of metrics, we focus on the quantitative metrics that measure how a companies’ carbon emissions align with Paris-compliant decarbonisation pathways. From a climate perspective, only actual emissions determine whether a Paris-compliant carbon budget is met.” (line 237-244) We have further shifted the framing of the article into three components – 1) conditions for PCP, 2) operationalisation requirements and 3) metrics to measure progress. We have moved the comparison with the TPI and ACT to the supplementary materials. The intention of our paper was to constructively establish rigour and overarching principles through which Compliance with the Paris goals could be interpreted rather forming a criticism of work to date.
Data availability and scope of this method: The proposed method requires quite a lot of detailed data about company emissions and it requires data about current and future market shares. These data are likely to be held by some companies, particularly those forward-looking companies with a clear climate strategy. Some of these data will be publicly disclosed, but some probably won’t be. As such, I venture that the proposed method would fit well with an approach like SBTi, which works via company collaboration. That is, companies looking to set a ‘science-based’ target that is accredited by SBTi apply the method themselves using their internal data and then turn to SBTi for accreditation. The wider world doesn’t see the inner workings, just the outcome (an accredited target).	Data availability: Data availability is a valid concern and is the main limitation of using the SDA method, compared to other methods, such as the Absolute Contraction method offered by SBTi, and even the GEVA or CSO method. This is because the SDA method scales the carbon budget to companies using production output/market share, whilst other methods such as GEVA and CSO only require gross profit (in addition to emissions, of course), which is more frequently available than production output, as for most publicly listed companies it is required to be reported. In addition, evaluation of climate performance is limited altogether for companies that do not disclose emissions, which is prevalent particularly for private companies.

I think it's important to contrast this with independent data of the sort that TPI provides. TPI does not work like SBTi, although it also uses the SDA. TPI assesses companies independently based on the data they publicly disclose. Most companies do not disclose the sorts of data required to implement the method proposed in this paper. There is rarely any visibility of company emissions at the facility/asset level. How do you make projections of future market share in a way that is consistent? Companies have strategic incentives to distort their market share projection under the SDA and more broadly. That is why TPI does not deal with changing market shares. This paper uses 10 Australian electricity utilities as a proof of concept, but developed-world electricity utilities do not reveal these problems, as electricity is the easiest of all sectors on which to apply the SDA. There is a lot of data available, because in countries like Australia, as I understand it, there is a good deal of mandatory reporting of energy consumption, emissions and so on. This is just not the case in almost any other sector. It is not the case in oil and gas, mining or any manufacturing sector I am aware of. The emissions data are often missing, or disclosed only on an intensity basis, or material emissions are missing, particularly in sectors like oil and gas where the larger part of the lifecycle carbon footprint is in Scope 3. I encourage the authors to reflect on these issues and what scope there is for application of this method. I would argue that the shortcuts taken by initiatives like TPI are a price to pay for trying to provide a large data set.

We have added this limitation to the main text: *“Further, company data availability is a concern in calculating PCP’s, especially for the SDA method as it requires production output in addition to emissions. We deliberately demonstrated PCPs with a data rich set of companies (Australian energy providers) and a data poor set (global cement companies) to demonstrate PCPs could still be adequately applied. Furthermore, the companies themselves can still make the necessary calculations to provide stakeholders interested in their Paris compliance, and our PCP approach provides directions to stakeholders on what data to request from companies.”* (line 370-377)

The purpose of our metrics is to evaluate companies independently, and thus provide data based on publicly available data, which is similar to TPI’s approach. In that sense, we can only provide as much data as the TPI can, and are limited to companies that have data since 2014 (in most cases, when carbon intensity is available, carbon emissions are also available and production can be inferred). We would like to note that even though the data may not be available, it does not mean the companies do not have it. Using our methodology they can calculate their degree of Paris compliance either for internal use or to publicly release. We aim to develop and release a tool that allows companies to calculate their Paris compliance, in addition to releasing the results of our PCP approach freely online based on publicly available information.

We would also like to re-iterate that the point we make is an appropriate application of the SDA, with metrics that allow for an independent evaluation of how companies are performing against the Paris goals and therefore we respectfully disagree that our method would fit well with an approach like the SBTi. As outlined in the paper, we argue that the SBTi currently does not apply the SDA method appropriately given it only works if all companies are evaluated against a common base year, consistent with the underlying pathway, which we now argue should be 2015 or prior, to be able to claim Paris compliance. We provide a more independent evaluation and process than the SBTi.

As for certain sectors disclosing more information than others – estimations can be made from production output, especially for the oil & gas and mining sectors. The sectors with high scope 3 emissions likely need different approaches, such as proposed by Rekker et al. (2018) for the fossil fuel sector. What we aim to do in this manuscript is lay a foundation of what requirements need to be met to be “Paris Compliant”. The manuscript has now been reframed to align with this aim. There may be different methods that can be adopted to this end, what is needed is transparency and legitimacy of what is measured. We explain this here:

“As several Paris-Compliant pathways will be available for any company to evaluate its performance against, there needs to be transparency in terms of the three conditions of Paris-Compliance, specifically, i.e., what allocation method is used, the base year of the pathway(s) of the underlying method, the scenario that this method is consistent with (e.g. IEA B2DS or the SSP1-1.9), and whether they encourage differentiated responsibilities” (line 163-167)

We believe that the lesser concern is the projections of future market share, as these always require assumptions – an inherent feature of the SDA method. What is most important is that estimated data is corrected as soon as real information is available:

“The first operationalisation requirement is that when applying a methodology that relies on the assumption of a future variable, e.g., projected market share, to calculate the emission reduction pathways and carbon budget, the allocation must be adjusted as soon as the information for the realised variable is available. For example, if market share was an input variable and was projected to calculate the carbon budget in advance, the carbon budget should be adjusted every time the market share differs from the projection.” (line 170-176)

And that companies compensate for any lack of action to ensure they stay within their carbon budget:

“The second operationalisation requirement is that if a company does not meet the emissions reductions of their PCP in any year, they must

	reconstruct a “re-alignment” PCP that adheres to all the conditions above while maintaining the company’s carbon budget. This will mean that any company which fails to reduce their emissions sufficiently in any year must increase their reductions in subsequent years to compensate. In short, if a company has not met its target, a “re-alignment” pathway should be defined to make up for the action deficit. Finally, whilst some methods use carbon intensity measures, it is important that corresponding absolute emission pathways and carbon budgets are always calculated (Krabbe et al., 2015; Faria and Labutong, 2019). Ultimately, the absolute cumulative emissions determine whether the Paris goals are met of holding warming to less than 2 degrees above pre-industrial levels.” (lines 177-186) We agree that the asset level data for the electric utilities is not available for other sectors, but that does not mean that projections cannot be inferred in other ways, such as done by Rekker et al. (2018) using different production projections. To address this comment regarding the problems of data availability and the lack of a contrasting sector we have also included a demonstration of the PCP approach, and the challenges associated it, to both the more data rich Australian energy sector and a data poor global cement sector (rich and poor in the sense of publicly available data) and highlight the different approach in the text: “Note that for other sectors with limited availability, we can project emissions using, for example, the geometric growth of emission intensity from 2014-2019, which we have done for our sample of cement companies in Figure 2 (Fig e, f). Companies themselves are encouraged to release projections of their emission pathways.” (lines 276-279) Finally, companies themselves will hopefully have access to the data necessary to assess their Paris Compliance through a Paris Compliant approach and having achieved this aim are more likely to be transparent about their data to demonstrate any such claim.
Updating: Emissions scenarios get regularly updated. How would the proposed method deal with this? I think it could get complicated, given that the centrepiece of	Updating: we have debated this point intensively, and in the end agreed that the base year is most important, more important than changes in the scenarios (that you refer to in your first point). In the updated manuscript we

the method is this correction for excess historical emissions. When emissions scenarios get updated, two things happen. First, there is a host of changes to things like technology costs, demand etc., which reflect an updated and improved understanding of where the world is heading. You absolutely want to incorporate those changes. Second, account is taken of emissions in recent years. Assuming a fixed carbon budget (n.b. this can be a moving target, given evolving scientific understanding), what then happens is that past emissions are higher, and future emissions have to be lower. So excess historical emissions get embedded in the scenarios themselves. Where does that leave the companies?

now discuss the base year in a lot more detail with it forming one of the two required conditions for Paris Compliance.

It is crucial to distinguish between the two points that have been raised within the reviewer's comment. As for the first, technology costs etc, this applies to methods using sectoral pathways only (i.e., the SDA method) and they could indeed miscalculate the possibility of one sector having a technological breakthrough or unexpected cost reduction, meaning other sectors should get a slightly larger budget. However, given the IEA B2DS is already a conservative interpretation of the Paris Agreement (50% chance of limiting to 1.75C temperature rise), and the risk of always updating to account for global inaction is arguably the greater risk, it is not meaningful to keep updating to new scenarios. We aim to understand how companies are performing against the Paris Goals, which include "pre-2020" mitigation. For the CSO method, or any method that does not rely on sectoral pathways, this problem does not present itself. However, the second point raised within this review comment remains for these methods, which is the changing of the carbon budget.

When discussing the carbon budget, we have to distinguish between two factors: the carbon budget changing through an increased understanding of the science, and the carbon budget changing because of time passing. This firstly highlights the importance of the base year and the specification of it when making calculations and doing comparisons.

Secondly, the goals of the Paris Agreement are clear regarding the warming goal. This does result in some scientific uncertainty as to the exact carbon budget that will achieve this goal. However, this uncertainty associated with our scientific knowledge of the exact carbon budget for the Paris goals is much more manageable than the uncertainty associated with continuously allowing companies to align themselves with many different base years, versions of scenarios, and budgets through time, which lacks accountability for prior inaction. If the carbon budget changes because of time passing, this is not a problem for the company adhering to the PCP as they would only

	have to compensate for their fixed budget since the base year. This is one of the main contributions of our manuscript and is explained here: “Second, the base year from which progress is measured should be set in 2015 or prior and needs to be consistent with the underlying decarbonisation pathway (of condition 1), consistent with the “enhanced action prior to 2020” section of the Durban Platform for Enhanced Action CP.21 (2015) referred to in the Paris Agreement (Paris Agreement, Article 4, Part 8) and the pathways outlined in the IPCC special report on 1.5C (IPCC, 2018). Base years have been shown to clearly have a significant influence on a company’s carbon budget (Faria and Labutong, 2019). If companies are not evaluated against a common, and ambitious, base year, it is not possible to compare their actions-to-date against climate goals or against the progress of other companies, and virtually impossible to ensure actions are on target with the Paris goals. Technically speaking there are and will continue to be global emission reduction pathways that are consistent with a “well-below” 2°C temperature goal, that start in 2016 or later. However, these pathways are requiring faster decarbonisation rates with every year of global inaction. More importantly allowing base years of 2016 or later either allows the global carbon budget to be exceeded or unfairly reduces the established carbon budget of those companies who have been taking appropriate action since (or before) the Paris Agreement was signed.” (line 112-127)
How data are used: A philosophical debate that runs through this paper is how much data to provide to investors, stakeholders etc., how much flexibility data-users are given to interpret the data, and how much we as researchers try to do the job for those users. I would argue that TPI does not really prevent users from doing similar calculations to those set out in this paper. TPI overlays company emissions pathways on low-carbon scenarios and investors can analyse those and interpret them in different ways. This is what we see in reality – a diversity of interpretations. You can take a company whose	How data are used: We thank the reviewer for encouraging us to reflect on this philosophical aspect. We believe that it is important, as academics, to progress independent research on defining what Paris Compliant pathways are and academic rigour on how these pathways should be operationalised to be meaningful. The metrics are suggestions on what progress indicators would be meaningful to stakeholders of companies, including investors. Acknowledging the complexity of multiple methodologies, it is important that each are critically examined by independent parties through academic

emissions intensity has been falling more slowly than a below 2C scenario in the past and you can require it to reduce its emissions faster in the future. That is doable using TPI data as set out in Figure 1, it just requires the user to do some calculations. I suspect the authors of this paper would argue that it is too much to ask your typical data-user to do this. They need help and therefore one should use the proposed method instead. I can accept this. Reasonable minds can disagree on it. But you do lose flexibility. I would encourage the authors to reflect on this issue of where the boundary lies between research production and use, and also whether the metrics they have developed would find wide application and under what circumstances. They are somewhat complex, too complex for a lot of investors I would argue.	rigour. We provide conditions that we argue ensure consistency with the goals of Paris Agreement. Whilst the TPI (or ACT for that matter) do(es) not prevent users from making similar calculations to those outlined in our manuscript, it is indirectly providing scores that are used by stakeholders to make decisions, and we believe therefore that it is very important they are critically examined. Also, using our methodology we can make further calculations for all companies with publicly available information, and provide the results to the public to enable them to make Paris Compliant decisions. Rapid decarbonisation requires meaningful data to be as readily and easily available as possible for users. With frameworks such as the TCFD increasingly encouraging companies to, amongst others, disclose targets and metrics on their transition performance, we provide guidance on what Paris Compliant metrics and targets are. We are also proposing an approach that can be explained to stakeholders, and with the reframing of our paper, provide flexibility in terms of methodologies as long as they comply with the conditions and requirements outlined in the manuscript. An important aspect we also promote is the transparency available for the stakeholders in terms of eligible PCPs: “As several Paris-Compliant pathways will be available for any company to evaluate its performance against, there needs to be transparency in terms of the three conditions of Paris-Compliance, specifically, i.e., what allocation method is used, the base year of the pathway(s) of the underlying method, the scenario that this method is consistent with (e.g. IEA B2DS or the SSP1-1.9), and whether they encourage differentiated responsibilities.” (lines 163-167) The operationalisation requirements ensure companies do not exceed their Paris-Compliant carbon budgets.
Companies’ historical responsibility: Analogous to the issue of national responsibility for historical emissions, to	Companies historical responsibility: Whilst it would help emission reduction greatly if companies internally accept responsibility for emission

what extent do the authors think that companies will accept that historical excess emissions should factor into their future emissions targets, given how few companies have set Paris-aligned targets even using the more accommodating methods of ACT/TPI? Although the base year of the IEA ETP scenarios is 2014, they were first published in 2017. So, for the first three years of the assessment period companies would have had no idea what they were going to be compared with.	reductions in general, regardless of the base year, changes mostly come through external pressures from governments, shareholders and other stakeholders. It is to those that we aim to provide independent information. Larry Fink in his famous annual letters to CEOs has increasingly focused on the need for companies to demonstrate how their operations align with the Paris Agreement, the most recent of which we have quoted as requiring companies to “disclose a business plan aligned with the goal of limiting global warming to well below 2°C, consistent with achieving net zero global greenhouse gas emissions by 2050 [18].” (lines 72-74) The recent Shell court case also demonstrates how companies can be held responsible for historic emissions by courts, as Shell’s contribution to historic emissions was one of the key reasons for ruling in favour of more ambitious carbon emission reduction targets for Shell. Thus, there is a legal risk for companies that do not consider their historic contributions to climate change. It is indeed a shortcoming that there is usually a lag between the start of a decarbonisation pathway and the publication date. Yet, the science on climate change and the first agreements on reducing global carbon emissions have been in place since the late 20th century. Companies that have been paying attention to prior IEA pathways and have already started to decarbonise are more likely to be aligned with the IEA B2DS. For example, Microsoft aims to be carbon negative by 2030 and compensate for all of its emissions since 1975 by 2050.
Minimum action case: What is the basis for the assumption in this case that the generating mix is 50% gas and 50% zero-carbon. Why not 60:40? Or 30:70? How would you construct a minimum action case in other sectors?	This pathway was exemplary only and has been discarded due to the questions it raised with our reviewers.
Minor comments	
It is a little confusing that the method is presented in terms of absolute emissions yet Figure 1 is presented in terms of emissions intensity.	The purpose of Figure 1 was to compare the PCP to how the TPI and ACT track performance of companies, whom both focus on intensity measures only.

	Indeed we argue: “Finally, whilst some methods use carbon intensity measures, it is important that corresponding absolute emission pathways and carbon budgets are always calculated (Krabbe et al., 2015; Faria and Labutong, 2019). Ultimately, the absolute cumulative emissions determine whether the Paris goals are met of holding warming to less than 2 degrees above pre-industrial levels.” (lines 183-186). We have now moved the old Figure 1 to the supplementary materials where we have also moved the comparison of PCP with ACT and TPI.
Line 47: since the Paris Agreement aims to limit global warming to well below 2C (with 1.5C as a stretch target), and well below 2C does not require reaching net zero by 2050 (this is a rule of thumb for 1.5C), it isn't strictly accurate to say the Paris Agreement requires net zero by 2050.	Thank you – this part of the statements has been removed and now reads: “peaking of global emissions as soon as possible, and holding the increase in the global average temperature to well below 2°C above pre-industrial levels and pursuing efforts to limit the temperature increase to 1.5°C above pre-industrial levels (Paris Agreement, Article 4, Part 1). (lines 109-112)
Line 74: although IEA scenarios happened to be used in the original implementation of the SDA, other scenarios could be used.	We agree, however the sentence remains correct that the SDA uses the IEA method and therefore we have left it unchanged.
Lines 139-141: per my comment above, I question which elements of a firm-specific carbon intensity could be calculated by an independent research programme using only public disclosures.	Please refer to our explanation above.
Lines 211-213: this is incorrect – TPI provides full data on companies' emissions pathways set against sectoral low-carbon scenarios on its online tool. The data were aggregated into a categorical indicator in our Nature Climate Change paper.	What we are trying to say is that whilst the online tool allows you to visually see the level of misalignment with the IEA B2DS sectoral pathway, it does not provide a quantitative metric (like our Metric 1). This is important for data-users who would like to grasp the (mis-) alignment of the company against a PCP without viewing the graph in the TPI tool.
It would help the exposition of the method to provide more intuition for what the formulae are doing, e.g. the PCP_e and PCP_i. Simon Dietz London 2nd Feb 2021	Thank you for the advice. We have replaced PCPe with PCP_{emissions} and PCPi with PCP_{intensity}.

Reviewer #3 (Remarks to the Author):	Response to Reviewer 3:
Main feedback: The paper proposes a methodology to benchmark and measure “transition performance” using a “strict science-based approach” and compares it to the benchmarks used in two initiatives that “have proposed methodologies to benchmark companies’ performances against science-based emission reduction levels”. In doing so the paper fails to acknowledge and provide context on two important factors:	We thank the reviewer for their insights and encouragement to focus our efforts more clearly on our concept of Paris Compliance which has been extremely helpful in the complete reframing of the manuscript. The specific changes are explained below.
1) first that these initiatives propose to measure and assess companies alignment to Paris goals on a multi-dimensional perspective; On the first point, the authors fail to acknowledge that, for example, the ACT initiative measures Paris-alignment based on a variety of quantitative and qualitative indicators. For the Electric Utilities sectors it proposes 17 quantitative indicators (only 3 of which respect to target setting, being the rest about Material investments, intangible investments, management, policy engagement and business model) and 5 qualitative indicators (Business model and strategy, Consistency and credibility, Reputation, Risk and Trend). The TPI also considers several other indicators related to management and management quality. Thus, the article does not acknowledge or discuss the multi-dimensionality of the initiatives it uses to compare its “Paris-aligned benchmark”, rather narrowly focuses on a “one-dimensional” approach to measuring Paris-alignment based on a company science-based targets and then making inferences from there which are not always valid (see further below).	This point is well taken. We agree that it was not clear from our paper that the TPI and ACT include a wide range of metrics to examine corporate action. We have addressed this firstly with a redirection of the paper to provide a broader assessment of the underlying allocation methods, including the SDA upon which SBTi, TPI and ACT are at least partially based. By doing so we have moved the focus away from TPI and ACT and on to the core focus of defining a set of conditions and operationalisation requirements for methodologies to provide advice on alignment to the Paris goals – what we are referring to as Paris Compliance. Secondly, we clarify in the scoping of the paper that we focus on quantitative metrics associated with these Paris goals, while acknowledging that qualitative metrics are also useful, particularly for evaluating the likelihood of quantitative goals being met (lines 223-227). Specifically, we state: “There are two recent initiatives that assess a company’s transition performance using the SDA method: 1) The Transition Pathway Initiative (TPI) [25]; and 2) Assessing low-Carbon Transitions (ACT) [26]. These two initiatives include a wide variety of qualitative and quantitative metrics to assess a companies’ performance. Acknowledging the importance of the variety of metrics, we focus on the quantitative metrics that measure how a companies’ carbon emissions align with Paris-compliant decarbonisation pathways. From a climate perspective, only actual emissions determine whether a Paris-compliant carbon budget is met.” (line 237-244)

	Although our approach may seem “one dimensional” from a climate change perspective, actual emissions are of primary importance to meeting these Paris goals, and hence we focus on how a company is performing in reference to these clear Paris goals (i.e., peaking of global emissions as soon as possible, and a warming of either 1.5 °C or well below 2 °C - Part 1 of Article 4 of Paris Agreement.)
2) that the premises on which it basis its “strict science-approach” might be “socially flawed”, due to the inherent difficulties of allocating carbon budgets on a top-down manner.	We acknowledge that the Paris Agreement was inspired by a bottom-up approach in which countries volunteered their NDC’s. However, our primary focus is on the Paris goals that this approach inspired - the goals of net-zero emissions, peaking of global emissions as soon as possible, and a warming of either 1.5°C or well below 2°C. As discussed above we allow for equity considerations to be volunteered in a Paris Compliant Pathway but we feel it necessary to require the carbon budget associated with the Paris goals to be strictly met otherwise the Paris goals become merely aspirational.
On what target indicators respect, the main assumption that distinguishes the approach taken in this paper is the decision to ensure that “each company strictly adheres to the Paris carbon budget”. This decision, which was explicitly avoided in the construction of ACT methodologies and science-based targets, is a momentous decision whose implications are not properly discussed in the paper, namely;	As discussed above we have addressed this specific comment regarding the ACT methodology in focussing the paper exclusively on the assessment of allocation methods on achieving the goal of well below 2 °C. We were unable to find any documentation justifying why the ACT decided on not strictly adhering to the Paris Carbon Budget so we were unable to contrast this approach directly with that of the ACT. We address each of the individual points explicitly below.
1) the basis and consequences of choosing any given base year for the budget allocation exercise (paper chosen 2014), as well as the implications of choosing any given scenario or science-based target setting method, particularly at the light of what is discussed in Faria and Labutong (2019)	This is a valid argument, but this is a clear point of difference in the approach we have taken. Our focus is on Paris Compliance in terms of achieving the well-below 2 degrees goal of the Paris Agreement, and we do not believe it is possible to claim the actions of a company are aligned with this Paris Agreement goal unless the conditions we have set out, including a set base year, are met. We have included a reference to Faria and Labutong (2019), which provides clear evidence and a demonstration of the importance of setting a common base year. Given the Paris Agreement was made in 2015 this year was considered the most appropriate base year, but

	we wanted to allow for companies that were most proactive and had demonstrated progress prior to that year and to accommodate the use of Paris aligned scenarios such as 2BDS used by the SDA, which has a 2014 baseline. We have specified in the text that “the base year from which progress is measured should be set in 2015 or prior and needs to be consistent with the underlying decarbonisation pathway (of condition 1), consistent with the “enhanced action prior to 2020” section of the Durban Platform for Enhanced Action CP.21 (2015) referred to in the Paris Agreement (Paris Agreement, Article 4, Part 8) and the pathways outlined in the IPCC special report on 1.5°C (IPCC, 2018). Base years have been shown to clearly have a significant influence on a company’s carbon budget (Faria and Labutong, 2019). If companies are not evaluated against a common, and ambitious, base year, it is not possible to compare their actions-to-date against climate goals or against the progress of other companies, and virtually impossible to ensure actions are on target with the Paris goals. Technically speaking there are and will continue to be global emission reduction pathways that are consistent with a “well-below” 2°C temperature goal, that start in 2016 or later. However, these pathways are requiring faster decarbonisation rates with every year of global inaction. More importantly allowing base years of 2016 or later either allows the global carbon budget to be exceeded or unfairly reduces the established carbon budget of those companies who have been taking appropriate action since (or before) the Paris Agreement was signed.” (Line 112-127)
2) The influence of the market share parameter in calculating any emissions budget forward and on what basis are the future projection of activity calculated and legitimized;	The future market share parameter is always going to be an assumption, whether these are projections by the company itself or estimated externally. This is an inherent feature of the SDA method. However, it is important that metrics are adjusted every year when the market share is known. This is one of the important adjustments that is not mentioned by the ACT methodology.

	We have identified this as an operationalisation requirement: “The first operationalisation requirement is that when applying a methodology that relies on the assumption of a future variable, e.g., projected market share, to calculate the emission reduction pathways and carbon budget, the allocation must be adjusted as soon as the information for the realised variable is available. For example, if market share was an input variable and was projected to calculate the carbon budget in advance, the carbon budget should be adjusted every time the market share differs from the projection.” (Lines 170-176)
3) How it would consider/not consider historical contributions and equity concerns within that allocation, a topic amply discussed in the scientific literature;	We agree that equity concerns are an extremely important topic and explicitly mentioned in the Paris articles. The Paris warming goals obviously have equity implications, particularly intergenerational equity, and a need to recognize that lower income groups are more vulnerable to climate change and less access to adaptation support. For this reason, we have also included an additional desired by not necessary equity condition in recognition of the UNFCCC principle of Common but Differentiated Responsibilities and Respective Capabilities (CBDR-RC) - Part 2 of Article 2 of Paris Agreement. However, we only include equity as a desirable condition primarily because it is a multi-dimensional complex issue including historical contributions of developed countries, disparity in current emission levels and current income levels, and intergenerational equity. As such it would not be easy to benchmark making it difficult to justify requiring it be included as a necessary condition.
4) how it would address carbon budget allocation for new companies and new assets;	We thank the reviewer for pointing out this oversight. This is indeed an important limitation in the application of allocation methodologies. We have included this as an operationalisation requirement with an example approach on lines 189-193. “In the case of new companies, a defensible approach must also be applied. For example, a new company has five years to establish their market share and emissions, following which they must then align their reductions to a

	constant reduction rate that achieves net zero as required by the chosen methodology for their sector.”
5) carbon budget re-allocation challenges when companies have significant changes within their corporate structures, e.g. mergers and acquisitions or sales of assets, namely: would that trigger a recalculation on a re-baselined 2014 intensity or from the year of the significant change; how that recalculation would be done; would the emission of the assets be carried to the new company carbon budget, or the old one;	This is another challenge that is not addressed in the SDA methodology itself. We thank you for bringing this to our attention. We have included guidelines for this on line 188-189: “In the case of a merger or acquisition the combined company must reduce its emissions as if both companies have been one company since the base year.”
6) the significant uncertainties around carbon budgets and the continuing evolution of scientific understanding of these and how changes in these could lead to changes in the proposed company carbon budgets under the proposed method.	Firstly, this would only need updating if the improved scientific understanding led to global budgets being higher than anticipated because the stated goal of well-below 2 degrees allows us to handle the situation where the estimated carbon budget was too small. Secondly, given the consistent and well documented alignment between warming and GHG emissions we believe it is reasonable to set the targets based on our current knowledge. We believe the uncertainty created by allowing leniency on our Paris Compliance conditions - that companies can continually change their base year and underlying scenarios - would be far greater than our scientific uncertainty around the relationship between GHG emissions and global warming. Therefore, we explicitly do not allow for the global carbon budget to be updated as such a significant update would likely require a new global agreement which could initiate its own set of new conditions, base years, and pathways.
Finally, there is no acknowledgement of the difficulty in doing this type of top-down allocation (beyond some of the issues flagged in 1 to 6 above) and how fundamentally contradictory it is in face of the spirit of “bottom-up” governance and top-down accountability proposed by the Paris agreement. A final note on this point to recognize that the ACT Framework and ACT methodologies are a multi-stakeholder process with strong representation of companies and where this approach would be very difficult to agree. The ACT initiative was launched at COP21 with the objective of creating higher	As mentioned previously we acknowledge that the Paris Agreement was inspired by a bottom-up approach in which countries volunteer their NDC’s. However, our primary focus is on the Paris goals that lie at the heart of the Paris Agreement - to hold warming within either 1.5 °C or well below 2 °C. This is what the Paris Agreement is known for and the association with the Paris Agreement that we argue stakeholders are looking for when they look for Paris Compliance. Therefore, our aim is to provide independent data for stakeholders to make decisions regarding whether companies are in

accountability and transparency on the multitude of company pledges to contribute towards Paris goals.	alignment with this Paris goal. This is a key difference with the ACT – we aim to provide academic rigour to what Paris Compliance means. We therefore also moved the comparison with the ACT to the supplementary materials as it is not our goal to criticise, but to propose an approach that is aligned with achieving the Paris goals. Current ratings on climate performance are mostly comparative to other companies and not to science-based limits (Rekker et al., 2019). This is what we are trying to address. What is the key purpose behind countries making NDC’s even if those combined NDC’s do not meet the goal? The ratcheting every 5 years to improve ambition to align with these goals is not explicitly stated in the Paris articles but it is mentioned a number of times with the purpose of ensuring the Paris goals can be met.
Addressing some issues of detail, the main criticism of ACT target indicators seems to be that they do not consider “past performance” but instead, at each assessment “recalculate” the company benchmark (CBg). Basically, the authors compare some ACT indicators on a basis of fixed budget allocation per company - when they are not supposed to operate in this way and doing it leads to distortions and invalid inferences and conclusions. The ACT indicators calculate, at each point in time an assessment is done (reporting year), a pathway for the company which is used as reference to measure some of its indicators - instead of comparing it with a previous and largely arbitrary fixed point in time.	First, we would like to clarify that the company budget is not fixed. It depends on the what variables are used for any projections that are included in the allocation methodology, we explain this on line 170-176: “The first operationalisation requirement is that when applying a methodology that relies on the assumption of a future variable, e.g., projected market share, to calculate the emission reduction pathways and carbon budget, the allocation must be adjusted as soon as the information for the realised variable is available. For example, if market share was an input variable and was projected to calculate the carbon budget in advance, the carbon budget should be adjusted every time the market share differs from the projection.” Second, we are looking at the problem from a different perspective to ACT. Our approach is concerned with ensuring companies are aligned with the Paris goals of well below 2 degrees. We are not saying that the ACT does not have excellent and well thought out components, but judged by this one standard alone the ACT approach is problematic. If all companies are allowed to continually recalculate their pathways in each new reporting year it is virtually impossible to ensure companies will be aligned with the

	carbon budget of the Paris Agreement - and hence the Paris carbon budget is unlikely to be met (Bjorn et al., 2021). This is the focus of our differentiation with other methods like ACT and what we demonstrate in the Supplementary Materials Figure 2. The Paris goals are a key focus of this manuscript and we have endeavoured to make this much clearer for readers, including in the new manuscript title. As discussed above maintaining a consistent base year across all companies is the only way to ensure the Paris goals will be achieved. The choice of 2015 is not arbitrary but rather chosen as the year the Paris Agreement was signed, ensures the Paris aligned scenarios developed by the IPCC and IEA can be used, and we allow for prior base years so that those companies that already begun decarbonising prior to this year to get credit for this previous effort.
The analysis done by the authors, assumes the scenarios and allocation mechanisms for companies will stay the same for companies, as time passes. Under this assumption new recalculated targets pathways are necessarily always more lenient and do not meet global carbon budgets – that is correct. However, ACT assessments do not assume fixed scenarios and fixed global carbon budgets in time. In fact, as carbon budgets get “consumed” and scenarios are revised, one’s expectation would be that these pathways will, necessarily, get more ambitious reflecting a smaller volume of the carbon budget, but also improved scientific understanding of carbon budgets. Adjustments for companies’ “under-achievements” – and overall economy under-achievement - should be reflected as the climate and social scientists reflect about the implication of those facts and how to reflect them in terms of future scenarios – and not on a basis of a carbon budget set arbitrarily at some point in time.	Like the previous comment, this is one of the main points we have now clarified in our manuscript in much more detail. Relying on companies to continually update their emission pathways based on updated scenarios and budgets is an impractical and potentially very confusing solution for achieving the Paris goals, e.g., keeping track of which version of scenarios and budgets companies are on. Such revaluations are unnecessary and risk inaction that is not fair to companies who have been taking action. We now specifically refer to this as follows: “Technically speaking there are and will continue to be global emission reduction pathways that are consistent with a “well-below” 2°C temperature goal, that start in 2016 or later. However, these pathways are requiring faster decarbonisation rates with every year of global inaction. More importantly allowing base years of 2016 or later either allows the global carbon budget to be exceeded or unfairly reduces the established carbon budget of those companies who have been taking appropriate action since (or before) the Paris Agreement was signed.” (lines 121-127)

	The goals of the Paris Agreement are clear regarding the warming limit. This results in some uncertainty in what carbon budget will achieve this limit. However, as stated above this uncertainty associated with our scientific knowledge of the exact carbon budget for the Paris goals is much more manageable than the uncertainty associated with continuously allowing companies to align themselves with many different base years, versions of scenarios, and budgets through time, that lack accountability for prior inaction.
One of the characteristics of the ACT Framework and its methodologies, is that it explicitly acknowledges that it can be “plugged-in” to a variety of scenarios: “From a more general point-of-view, sector benchmarks may be changed from the 2DS scenario to another relevant low-carbon scenario depending on the availability of geography-specific, context-specific, future relevant scenarios, or future updates of the 2DS scenario – and possibly to explore higher ambition requirements. In this case, the ACT assessment report shall disclose which low-carbon background scenario has been used. This also adds some flexibility to the assessment of specific companies and to the use of the most up-to-date or more ambitious (e.g. IEA ETP B2DS) scenarios.” (page 17, ACT Framework). ACT EU benchmarks were first utilized in 2016, when only the 2DS was available. The current version of the methodology (1.1, published in early 2019) is basically an edited version for a common ACT visual layout of the 2016 methodology. This does not reflect the fact that in more recent assessments, like the ones done for the World Benchmarking Alliance, the IEA ETP B2DS has been used as reference scenario (see https://www.worldbenchmarkingalliance.org/publication/electric-utilities/methodology/) . Under a perspective of a fixed-based year and set carbon budget, it is true that target recalculations under same scenario and for companies that have complied with Paris-compliant trajectories would lead to over-allocation. However, if you do not follow that constraint – as ACT does not - the expectation is that the global carbon budget constraint can still be met as scenarios get updated, reflecting both changes in the understanding of global carbon budgets (and their significant uncertainties)	We have removed the comment that the ACT does not use IEA B2DS in the Supplementary Materials, where most of the comparison with the ACT and TPI has been moved. We hope that we have adequately addressed the other points regarding changing scenarios and budgets in our responses above.

as well as the “consumption” of the carbon budget by economic actors. Companies’ new benchmark trajectories will be more ambitious than previous trajectories – achieving the same effect as the PCP is proposing to do, but in a different way.	
Detail feedback: The authors’ state “The ACT also strays from the most recent SDA approach in using the IEA 2DS pathway instead of the IEA B2DS pathway. Even if the ACT were to evaluate all companies’ performance since 2014 using IEA B2DS (as illustrated for the company in Figure 1b), their “previous” benchmark does not adjust for discrepancies between the actual and projected market share.” We do not totally follow what is meant by this sentence. ACT does not make a “previous” benchmark. It has specific indicators to assess if companies’ past targets have been reached or not (EU 1.3 and EU2.1 and to a certain extent EU 2.2), which fit its different philosophy on how to assess Paris-alignment of companies.	In the ACT General Methodology (2019) the “Company’s previous benchmark” (see Supplementary Figure 1) is calculated using projections of market share. Let’s consider a case where the projection of market share was higher than projected, this would mean the carbon intensity target would have had to have been lower if the company had accurately predicted this. Now let’s say the company has followed this originally set target, and thus could get a score that says they reached their target. Is this accurate? No, because the target was not accurate – it was not based on actual market share. This is the point we are making. When assessing if a target, or an emission reduction pathway, has been met, the target must be adjusted for real information when it becomes available. This is one of the key operationalisation requirements we explained above.
The supplementary material is more specific and it flags as limitations “It is unspecified what year “initial” refers to, and even if it was used at a common base year consistent with the IEA pathway, it is unclear if and how the benchmark is adjusted to account for performance to date, especially how the benchmark needs to be adjusted to make up for any deficits. The ACT Framework uses “previous” and “current” benchmarks which at least provide clarity on what year the initial intensity it is based on, but here it is unclear.” “Initial” refers to the “reporting year” this is, the latest year for which data is available to assess the specific indicator. The method is not explicit, but as the authors correctly inferred, there is no “adjustments for performance to date” at the level of this specific indicator – there is a specific indicator for that purpose, as mentioned above.	Thank you for the clarification. We see that this has been corrected in the latest version of the ACT documentation and so we have removed these comments from the Supplementary Materials.
While the author's characterize how the proposed indicators are superior to existent ones there is no critique of the proposed indicators. Should we conclude that this is the perfect set of indicators to measure company	As discussed above our key point of difference, as has been made clear by the comments of our reviewers, is that we are focussed on assessing whether companies are compliant with the Paris warming goals. In this regard we do believe we show clearly how the PCP and metrics will perform better at this

alignment with Paris goals according to authors or are there any limitations to its use also?

key task. As the reviewer points out there are indeed limitations to our method which we discuss in lines 361-377:

“We used the SDA method to demonstrate the PCP approach. It should be noted that the SDA method itself also has some known limitations within its foundations (Bjorn et al., 2021), such as allowing companies with a higher initial intensity to have a greater share of the remaining global carbon budget (i.e., grandfathering). In addition, the IEA B2DS pathway used by the SDA relies on significant amounts of carbon capture and storage and negative emissions technologies which are currently not being deployed at anywhere near the rate required by this scenario [21, 34], which does not alter the total carbon budget but could have implications for the sectoral breakup of the budget. The B2DS pathway is also only consistent with the more conservative interpretation of the Paris Agreement with a 50% chance of keeping global warming below 1.75°C. Further, company data availability is a concern in calculating PCP’s, especially for the SDA method as it requires production output in addition to emissions. We deliberately demonstrated PCPs with a data rich set of companies (Australian energy providers) and a data poor set (global cement companies) to demonstrate PCPs could still be adequately applied. Furthermore, the companies themselves can still make the necessary calculations to provide stakeholders interested in their Paris compliance, and our PCP approach provides directions to stakeholders on what data to request from companies.”

ACT context

Considering that ACT initiative is taken as an important point of comparison for the article, it is important that it is properly characterized. As such, the article should address the following points:
“ACT is a voluntary initiative of the UNFCCC secretariat Global Climate Agenda supporting corporate climate accountability. It develops sectoral methodologies through a multi-stakeholder process, as an accountability framework to support companies with delivering low carbon transition strategies and actions aligned with the Paris Agreement mitigation goal. ACT was launched in 2015 at COP21 (Paris) and released its ACT Framework, first 3 methodologies and benchmarks in 2016 at COP22 (Marrakesh).”

We thank the reviewer for the comment and agree that we have inadequately presented the many valuable contributions of the ACT. We have updated the introduction to the ACT in Supplementary Material using some of the reviewers points directly:

“The ACT methodology, developed in 2019, defines benchmarks based on a company’s carbon intensity and market share. The ACT is a voluntary initiative of the UNFCCC secretariat Global Climate Agenda supporting corporate climate accountability. It develops sectoral methodologies through a multi-stakeholder process, as an accountability framework to support companies with delivering low carbon transition strategies and actions aligned with the Paris Agreement mitigation goal. As such it is more

aligned with the spirit of “bottom-up” governance than the top-down accountability proposed by the Paris agreement goals. It is also a highly flexible approach that can be “plugged in” to a variety of scenarios.

Given its design and purpose it is somewhat unfair to critique the ACT approach based on the one-dimensional warming goals of the Paris Agreement. This is however the key focus of the PCP approach and in this regard the ACT approach presents some problems.” (Supplementary Materials, line 37-47)

Reviewer comments, third round

Reviewer #1 (Remarks to the Author):

Thank you for revising your study. I think you have done a good job at addressing my review comments and that your reframing of the study increases its relevance. I have three main comments and a bunch of minor comments.

First, the draft suffers from some unfortunate formatting issues. Figure 1 seems to not be displayed correctly with the legend referring to two panels (a and b), while there only appears to be a single panel. Likewise, line 260 refers to a Figure 1c that does not appear to exist. Perhaps the issue relates to overlaid images? The reference list contains several references with the same number, the sequence is odd, and there seems to be a general misalignment between the numbers cited in the body text and the numbered references in the list. These formatting errors makes it difficult to assess parts of your study, but should be easy to fix.

Second, I think your proposed criterion related to a baseline year of 2015 or earlier requires some elaboration. Are you proposing a common base year (applicable to all companies)? It sounds like that around line 116, but it is not clear. If so, I encourage you to make it explicit, since companies commonly chose their own base year when setting emissions targets, also within the SBTi scheme. Also, please elaborate on why you find 2015 (or earlier) to be a desirable base year. Your brief reference to the "enhanced action prior to 2020" section of the Durban Platform for Enhanced Action CP.21 is not sufficient for me to understand this. See more detailed questions related to the base year below.

Third, I think that the transition between the conditions (and additional operational requirements) and the presentation of your metrics, around lines 207-243, can be improved. Currently, you present the TPI and ACT approaches as part of the transition (lines 232-243), but this presentation appears a bit rushed since it is not fully explained what makes these existing approaches inadequate. Perhaps it would work better if you moved the references to TPI and ACT to the discussion section (keeping the main assessment of these approaches in the supplement). In doing so, your transition could focus on the need for metrics when evaluating companies' transition performance relative to PCPs.

Detailed comments:

- Line 107: Perhaps here would be a good place to refer to the SBTi foundations for target-setting document (as you do in Table 1), since it offers a way to operationalize the "as soon as possible" criterion.
- Line 116: How can a base year be ambitious? Does it relate to the assumed timing of peak emissions?
- Line 122-124: Not sure I understand. Any common base year will be "unfair" to companies that had done a lot to reduce emissions in the time leading up to it, right?
- Table 1: Should "they" be "SBTi" under condition 2 for ACA?
- Line 190-193: I am not sure I agree with such an arbitrary threshold under which a company is automatically "Paris Compliant". In some cases, even small companies might want to check their alignments against a Paris pathway (as evidenced by the SMEs that have set SBTs).
- Line 272: Should it be "limited data availability"?
- Line 316: Should it be "before 2040" (looks like around 2039 for company #6 on the figure)?

Reviewer #2 (Remarks to the Author):

The authors have made major changes to the paper in response to my comments and I think these have helped to position it more appropriately and make it more solid. I was hoping to waive it through at this stage, but there are some concerns that I can't quite shake off. Also, there are some problems with the formatting and presentation of the manuscript, which mean I can't

properly evaluate it.

Major comments

Formatting issues

Figure 1 panel (a) seems to be missing from the article. This figure plays an important role in understanding the methodology. Also, the cement companies in figure 2 are not listed, unless I missed this somewhere.

Structure/presentation

I think it was a good move to refocus the paper around a set of principles or conditions that methodologies need to meet and that companies need to meet when implementing the methodology. But the sum of these conditions, plus some auxiliary technical conditions, makes for a rather complicated manuscript. My suggestion is that the authors create a schematic/table that summarises them all and make this the visual centre-piece of the paper.

Historical responsibility

I think my review hinted at this before (and I notice another reviewer did too) – holding companies accountable for their historical emissions is a key principle underpinning this paper's proposals. It is a perfectly defensible principle, but it is debatable, especially in a context where essentially the whole global economy (almost all countries, almost all companies) has been exceeding its Paris-compliant pathway since 2015. I certainly don't want to block the paper on this issue, but I think the fairness/justice underpinnings of this condition need to be more explicitly disentangled from the purely physical/scientific underpinnings of condition #1. At the moment, you could read the paper as saying it is scientifically required that companies compensate for their past exceedance, whereas in fact we could – I'm not saying this is 'right' – just press reset, rebase emissions, and calculate companies' future pathways afresh.

Base year and scenarios becoming out-of-date

This issue is still of concern to me. Apologies if I am being obtuse but I will try to (slightly) restate the problems as I see them. When a modelling organisation like IEA updates its scenarios, (i) the base year shifts forward (from 2014 for the ETP2017 scenarios like B2DS to 2018 or 2019 for the more recent SDS scenario that is basically an update of B2DS), (ii) the emissions estimate for 2018/19 gets revised due to recalibration/rebasing, and (iii) the future pathway changes. Given (i)-(iii), I still struggle to see how companies' historic over-emissions can be calculated at the same time as ensuring their future pathways are Paris-compliant according to the latest understanding of costs and so on. Can you explain how to use a fixed base year while updating future pathways? Will we be stuck using ETP2017? It is already looking out-of-date. The response document argues that "it is not meaningful to keep updating to new scenarios", so this may give the answer. I'm afraid I would find that quite a hard position to defend. The passage of time will show that mid-2010s low-carbon scenarios were wrong in many respects (e.g., costs of solar, costs/feasibility of CCS). It's hard to imagine sticking with them.

Minor comments

Paris and net zero

I think more precision is needed on what Article 2 of the Paris Agreement requires for emissions. As I understand it, "well below 2C", defined as say 1.75C, does not require net zero emissions by around 2050, it is 1.5C specifically that does. And 1.5C requires net zero emissions of CO₂ but not of other GHGs, which can be subject to positive but decreasing emissions at that time, a seemingly small point that is nonetheless getting actors in the real world into a mess about, for example, agribusiness and countries like New Zealand with large agriculture sectors.

Updating market share

What is a good way of updating the carbon budget when new information becomes available about variables like market share? Is market share relatively stable over time or is it volatile like revenue? If it is volatile, should we let companies' goalposts be similarly volatile, or should longer-term projections be more stable? These are genuine – non-rhetorical – questions. Bayesian updating might work here.

Figure 1

I assume the measure in panel b is *cumulative* CO2 emissions. I think this should be explicit.

POINT BY POINT RESPONSE TO REVIEWER COMMENTS

Reviewer #1 (Remarks to the Author):	Response to Reviewer 1:
Thank you for revising your study. I think you have done a good job at addressing my review comments and that your reframing of the study increases its relevance. I have three main comments and a bunch of minor comments.	We thank you for reviewing our manuscript again and are pleased you think we did a good job in addressing your previous comments. Below we respond in detail to your comments.
First, the draft suffers from some unfortunate formatting issues. Figure 1 seems to not be displayed correctly with the legend referring to two panels (a and b), while there only appears to be a single panel. Likewise, line 260 refers to a Figure 1c that does not appear to exist. Perhaps the issue relates to overlaid images? The reference list contains several references with the same number, the sequence is odd, and there seems to be a general misalignment between the numbers cited in the body text and the numbered references in the list. These formatting errors makes it difficult to assess parts of your study, but should be easy to fix.	We apologise for this formatting issue. This should be displaying correctly now. We submitted two Word versions, one with tracked changes and one without. We spent several hours trying to display the yellow line correctly (somehow it kept disappearing when converting to PDF). In the very final submission, something must have gone wrong, apologies again. We have also changed the yellow line to a purple line, which does not seem to have the same issue. Figure 1c has been corrected to 2a (we added a new Figure 1, and thus it is now Figure 2a). The reference list is automatically numbered and thus we couldn't identify any references with the same number in the reference list itself. For the in-text referencing, when reshuffling text we had indeed made some errors and they have now been fixed. Note that sometimes a prior reference is used again, and then the numbering is not sequential. For example, in line 77, we refer to reference 13, 20, 21, and 22. Reference number 13 was already mentioned prior in line 63. Other than those instances, the numbering is now sequential.
Second, I think your proposed criterion related to a baseline year of 2015 or earlier requires some elaboration. Are you proposing a common base year (applicable to all companies)? It sounds like that around line 116, but it is not clear. If so, I encourage you to make it explicit, since companies commonly chose their own base year when setting emissions targets, also within the SBTi scheme. Also, please elaborate on why you find 2015 (or	We agree that the context of the base year could have been elaborated further, as the conditions now constitute a major component of the arguments for the paper. We have now further elaborated the reason for the baseline of 2015 or earlier: "We propose a base year of 2015 or prior to reflect the year in

earlier) to be a desirable base year. Your brief reference to the “enhanced action prior to 2020” section of the Durban Platform for Enhanced Action CP.21 is not sufficient for me to understand this. See more detailed questions related to the base year below.

which the Paris Agreement was signed and to capture emissions reductions that have been achieved well before 2020. This is also consistent with the “enhanced action prior to 2020” section of the Durban Platform for Enhanced Action CP.21 (2015) referred to in the Paris Agreement (Paris Agreement, Article 4, Part 8) and the pathways outlined in the IPCC special report on 1.5°C” (line 120-124)

A common base year is necessary to compare companies’ climate actions, which we now state more explicitly “Base years have been shown to clearly have a significant influence on a company’s carbon budget [29]. Further, we argue that the same base year needs to be used to fairly compare the performance of companies. Currently, the Science Based Targets initiative allows companies to choose their own base year. For example, a company could set a base year of 2020 for a target that compares their emissions reductions against a pathway (IEA B2DS) which has a base year of 2014, thus neglecting emissions between 2014-2020.” (line 124-130)

We further also clarify the different elements of condition 2: “Note that the need for a company’s base year to be consistent with the year of the underlying pathway is a purely mathematical argument – it is required to ensure the world meets the carbon budget of the Paris goals. But requiring that decarbonisation pathways start in 2015 or prior, is an ethical and fairness condition; to keep companies accountable for past emissions, and to ensure companies are compared fairly with their peers. Not accounting for emissions since the beginning of the Paris Agreement makes meeting the Paris goals more and more impractical, and infeasible. Besides, companies are likely to reduce their financial risk by meeting the second condition as such companies will be exposed to less transition risks than counterparts who delay action, part of the reason some stakeholders want to know whether a company is Paris Compliant.” (line 141-151)

Third, I think that the transition between the conditions (and additional operational requirements) and the presentation of your metrics, around lines 207-243, can be improved. Currently, you present the TPI and ACT

After re-reading this part, we agree the references to TPI and ACT were organised in a confusing way. After moving this reference to the discussion, we found it hard to find a place where it would flow well. We have given it

approaches as part of the transition (lines 232-243), but this presentation appears a bit rushed since it is not fully explained what makes these existing approaches inadequate. Perhaps it would work better if you moved the references to TPI and ACT to the discussion section (keeping the main assessment of these approaches in the supplement). In doing so, your transition could focus on the need for metrics when evaluating companies' transition performance relative to PCPs.	some thought and believe it is mostly solved by keeping it as part of the transition but placing it just before the “Metrics” section, rather than as part of its introduction. We have reduced the reference in size, and instead focused on explaining the need for metrics: “Defining conditions and operationalisation requirements for PCPs is not sufficient to evaluate how companies are performing against a PCP. For stakeholders to factor corporate climate performance into their decisions, the extent of their (mis-) alignment with the Paris goals needs to be quantified through metrics. There are two recent initiatives that aim to do this and assess a company’s transition performance using the SDA method: 1) The Transition Pathway Initiative (TPI) [32, 33]; and 2) Assessing low-Carbon Transitions (ACT) [34]. While both initiatives have done an excellent job on engaging with companies on emission reduction pathways, they both inherit shortcomings from the SBTi, with additional issues of their own, which we outline in more detail in the Supplementary Materials. We therefore propose here a set of new metrics to evaluate a company’s performance against PCPs.” (line 265-274)
Detailed comments:	
 • Line 107: Perhaps here would be a good place to refer to the SBTi foundations for target-setting document (as you do in Table 1), since it offers a way to operationalize the “as soon as possible” criterion. 	Thank you, we have added this reference in that place (now line 113).
 • Line 116: How can a base year be ambitious? Does it relate to the assumed timing of peak emissions? 	Thank you for highlighting that this needs more explanation. Generally speaking, the earlier the base year the more ambitious it is considered to be. This is because the earlier the base year, the more historic emissions the companies are held accountable for, which would keep them consistent with pathways that have earlier peaking of emissions, illustrating that the later the peaking of emissions, the more inconsistent there are with these pathways. To clarify we have amended as follows: “Our conditions focus on the need for a common, and early as practicable, base year for all companies and consistency with an underlying Paris-aligned decarbonisation pathway” (line 28-30)

	“If companies are not evaluated against a common, and as early as practicable, base year, it is not possible to compare their actions-to-date against climate goals and it becomes virtually impossible to ensure actions are on target with the Paris goals.” (line 130-133) We hope we have clarified this by addressing your earlier comment on base year above.
• Line 122-124: Not sure I understand. Any common base year will be “unfair” to companies that had done a lot to reduce emissions in the time leading up to it, right?	This is correct. In the manuscript, we focus on what a Paris Compliant company is, and propose that companies should -at least- be aligning themselves with a decarbonisation pathway starting in 2015 or prior. Whilst it is true from a compliance perspective, that an earlier date might seem unfair, ultimately a company will benefit from early action by reducing the need for more aggressive reductions later in the transition. There is nothing preventing a forward-thinking company from complying with even earlier decarbonisation pathways than those starting in 2015. However, we set a “latest” start date of 2015 as a line must be drawn to meet the Paris goals (as explained in the text) and the year the Paris Agreement was made seems a logical choice for “Paris Compliance”.
• Table 1: Should “they” be “SBTi” under condition 2 for ACA?	Yes, thank you for pointing this out. This has now been corrected in the manuscript.
• Line 190-193: I am not sure I agree with such an arbitrary threshold under which a company is automatically “Paris Compliant”. In some cases, even small companies might want to check their alignments against a Paris pathway (as evidenced by the SMEs that have set SBTs).	The reason this threshold was put in was mostly for new clean companies. For example, if there is an electricity generator that has mostly renewables but a bit of natural gas (to firm their supply), then it would be well on track. However, because their initial emission intensity is very low, they will only get a very small, and even negative carbon budget (as the sector ought to go negative under the B2DS). At hindsight, a company like we describe would be able to demonstrate their carbon intensity is well ahead of the sector and would for example comply

	with the EU taxonomy to be classified as green (i.e. less than 100g kWh/CO₂). We therefore have decided to remove the third condition.
 • Line 272: Should it be “limited data availability”? 	Yes, thank you for picking this up. We have corrected this in the manuscript (line 307).
 • Line 316: Should it be “before 2040” (looks like around 2039 for company #6 on the figure)? 	Yes, it should have read “all but one company is expected”, but we have changed it in accordance with your suggestion (line 357).

Reviewer #2 (Remarks to the Author):	Response to reviewer 2:
The authors have made major changes to the paper in response to my comments and I think these have helped to position it more appropriately and make it more solid. I was hoping to waive it through at this stage, but there are some concerns that I can't quite shake off. Also, there are some problems with the formatting and presentation of the manuscript, which mean I can't properly evaluate it.	We are happy to read the reviewer feels our revision has helped position the manuscript better and has made it more solid. We would like to thank the reviewer for their previous comments which have already helped improve our manuscript greatly. The comments raised by the reviewer are certainly valid and we address them in detail below.
Major comments Formatting issues Figure 1 panel (a) seems to be missing from the article. This figure plays an important role in understanding the methodology. Also, the cement companies in figure 2 are not listed, unless I missed this somewhere.	We sincerely apologise for this formatting error, it should be displaying correctly now. It seems to be an issue for non-mac systems – we noticed in a windows system would not show the yellow line, not even when converting to PDF. We thought this was fixed when it displayed correctly on the PDF (after trying many things over many hours), but we now notice it does not. We have now checked across multiple systems and it should display correctly. We have also changed the yellow line to purple which helped resolve the issue. Not listing the companies is an oversight that has been corrected (line 383-384).
Structure/presentation I think it was a good move to refocus the paper around a set of principles or conditions that methodologies need to meet and that companies need to meet when implementing the methodology. But the sum of these conditions, plus some auxiliary technical conditions, makes for a rather complicated manuscript. My suggestion is that the authors create a schematic/table that summarises them all and make this the visual centre-piece of the paper.	We are pleased you think it was a good move to refocus our paper. Thank you for the suggestion on how to make the manuscript easier to follow. We have now included a new figure (Figure 1) that illustrates the compliance conditions and operational requirements to set a Paris Compliant Pathway. We have included this early in the manuscript as we think the suggested Figure does make the methodology appear clearer and more straight forward to follow.

Historical responsibility

I think my review hinted at this before (and I notice another reviewer did too) – holding companies accountable for their historical emissions is a key principle underpinning this paper’s proposals. It is a perfectly defensible principle, but it is debatable, especially in a context where essentially the whole global economy (almost all countries, almost all companies) has been exceeding its Paris-compliant pathway since 2015. I certainly don’t want to block the paper on this issue, but I think the fairness/justice underpinnings of this condition need to be more explicitly disentangled from the purely physical/scientific underpinnings of condition #1. At the moment, you could read the paper as saying it is scientifically required that companies compensate for their past exceedance, whereas in fact we could – I’m not saying this is ‘right’ – just press reset, rebase emissions, and calculate companies’ future pathways afresh.

We appreciate your feedback which continues to improve our paper and helps to clarify our arguments. We have amended our paper to further explore the issues you raise in this comment.

Condition 2 has both a “mathematical” component, that the base year should align with the underlying decarbonisation pathway, and a “fairness” component (when comparing different companies), that only pathways from 2015 or prior be used to account for historic emissions.

In terms of aligning with a certain decarbonisation pathway, it is mathematically necessary to have the same base year as the decarbonisation pathways underpinning the compliance. For example, for the IEA B2DS, the SBTi is allowing a base year that is different to the initial year of the IEA B2DS, i.e. 2014. We have added a comment to this affect in the text: *“Base years have been shown to clearly have a significant influence on a company’s carbon budget [29]. Further, we argue that the same base year needs to be used to fairly compare the performance of companies. Currently, the Science Based Targets initiative allows companies to choose their own base year. For example, a company could set a base year of 2020 for a target that compares their emissions reductions against a pathway (IEA B2DS) which has a base year of 2014, thus neglecting emissions between 2014-2020.”* (line 124-130)

It is correct that we could just press reset, rebase and let companies start afresh, however if this happens repeatedly, as effectively has happened globally over the last 30 years, decarbonization pathways just become increasingly infeasible, putting climate goals further and further out of reach.

We have added the following sentences to clarify: *“Note that the need for a company’s base year to be consistent with the year of the underlying pathway is a purely mathematical argument – it is required to ensure the world meets the carbon budget of the Paris goals. But requiring that*

	decarbonisation pathways start in 2015 or prior, is an ethical and fairness condition; to keep companies accountable for past emissions, and to ensure companies are compared fairly with their peers. Not accounting for emissions since the beginning of the Paris Agreement makes meeting the Paris goals more and more impractical, and infeasible. Besides, companies are likely to reduce their financial risk by meeting the second condition as such companies will be exposed to less transition risks than counterparts who delay action, part of the reason some stakeholders want to know whether a company is Paris Compliant.” (line 141-151) And in the third condition we refer to it as well: “This means that companies in developed nations have a greater responsibility in mitigating climate change than companies in developing countries due primarily to their greater historic contributions to climate change and mitigation capabilities [1]. Our second condition loosely touches on accounting for historical contributions, but only to 2015 or prior and the underlying pathways may not have distinguished sufficiently between developed and developing nations.” (Line 154-160)
Base year and scenarios becoming out-of-date This issue is still of concern to me. Apologies if I am being obtuse but I will try to (slightly) restate the problems as I see them. When a modelling organisation like IEA updates its scenarios, (i) the base year shifts forward (from 2014 for the ETP2017 scenarios like B2DS to 2018 or 2019 for the more recent SDS scenario that is basically an update of B2DS), (ii) the emissions estimate for 2018/19 gets revised due to recalibration/rebasing, and (iii) the future pathway changes. Given (i)-(iii), I still struggle to see how companies’ historic over-emissions can be calculated at the same time as ensuring their future pathways are Paris-compliant according to the latest understanding of costs and so on. Can you explain how to use a fixed base year while updating future pathways? Will we be stuck using ETP2017? It is already looking out-of-date. The response document argues that “it is not meaningful to keep updating to new scenarios”, so this may give the answer. I’m afraid I would find that quite a hard position to defend. The passage of	We thank the reviewer for these comments. Point (iii) is arguably the most difficult challenge to overcome in defining sectoral decarbonisation pathways based on models – be they from the IEA or based on any other integrated assessment or alternate modelling framework. No model offers a forecast, just a plausible possible pathway under a given set of assumptions, many of which are uncertain and will be out-of-date soon after the model is run. Furthermore, no model has sufficient resolution of transparency to support perfection in any Paris Agreement compliance framework. However, it is only fair, that a compliant company that has met the emissions reductions required under an agreed compliant pathway at the date of the reset should be credited with those reductions, and that any steepening of the emissions reduction trajectory for the sector arising from underperformance by others, should be the responsibility of those

time will show that mid-2010s low-carbon scenarios were wrong in many respects (e.g., costs of solar, costs/feasibility of CCS). It's hard to imagine sticking with them.

underperforming peers. Otherwise, we are arguably penalizing early-movers to compensate laggards.

We have added a sentence on this point:

“There are multiple pathways that could meet this requirement, and both the choice of pathway and its underlying assumptions must be transparent.”
(Line 115-117)

And a whole paragraph in the conclusion outlining the limitations of our method:

“On a cautionary note, while our method holds companies more accountable to decarbonization commitments, it is not without its limitations, such as the limited number of modelled pathways that meet our strict criteria. Companies might reasonably prefer to align their PCP with a modelled pathway other than IEA B2DS. Given the plethora of macro-scale energy transition and integrated assessment models (IAMs) available, it may well be that specific models are perceived to offer a higher level of resolution and/or more up-to-date assumptions for their sector and region. However, with different companies in different sectors variously aligning themselves with different decarbonization pathways, it is likely that the aggregated company carbon budgets would be quite different to the budget if all companies were not aligned to a common path. This is especially problematic because of the number of IAM scenarios that allow significant overshoot of carbon budget followed engineered drawdown of CO₂ later in the century. This overshoot issue is particularly troubling, and one that deserves more attention by researchers and institutions considering the alignment of firms and sectors with the Paris Agreement.” (line 445-458)

Points (i) and (ii) are due to the shifting of the base year, and account for “inaction” or emissions between the prior scenario and the current (e.g. 2014 ETP 2017 and 2018 IEA SDS). This is not due to a new understanding of the science in terms of the overall global carbon budget (except for it having been “spent”). The scenarios with baselines 2015 or prior provide an adequate representation of the carbon budget. Changes in technology costs

	of solar vs CCS do not change the carbon budget. Technological advances could change what is considered a fair allocation of the budget between sectors, but this is a marginal correction. The greater goal is Paris Compliance and for that a line must be drawn somewhere, and soon, if the budgets are to be met. Please see our response to the previous comment for an explanation of the risk for global climate action if we keep pressing “reset” by allowing a new base year.
Minor comments Paris and net zero I think more precision is needed on what Article 2 of the Paris Agreement requires for emissions. As I understand it , “well below 2C”, defined as say 1.75C, does not require net zero emissions by around 2050, it is 1.5C specifically that does. And 1.5C requires net zero emissions of CO₂ but not of other GHGs, which can be subject to positive but decreasing emissions at that time, a seemingly small point that is nonetheless getting actors in the real world into a mess about, for example, agribusiness and countries like New Zealand with large agriculture sectors.	We agree that clarity of intent is essential. In this regard it is important to acknowledge the many assumptions, deep uncertainties, variations in modelled emissions trajectories (including whether carbon budget overshoot is allowed or not), and different climate impacts that are represented in the many scenarios produced by numerous different models that informed the Paris Agreement. We have rephrased this as follows: “Stabilising global temperatures in line with this primary objective of the Paris Agreement requires net greenhouse gas emissions to be reduced at unprecedented rates, with CO₂ emissions needing to reach net-zero by around mid-century (to stay within 1.5 °C warming levels).” (line 43-46) We shortened the reference in from Larry Fink on line68-69 to exclude the part where he incorrectly states the requirement for net-zero greenhouse gas emissions by 2050. Also, on page 13 we now specify that the net-zero applied to the sector and the IEA B2DS scenario: “around 2050 for the electric utilities sector in the IEA B2DS scenario).” (line 289-290)
Updating market share What is a good way of updating the carbon budget when new information	Estimations of production, on which market share is based, may vary by industry in their ease of prediction and is generally difficult to obtain. The

becomes available about variables like market share? Is market share relatively stable over time or is it volatile like revenue? If it is volatile, should we let companies' goalposts be similarly volatile, or should longer-term projections be more stable? These are genuine – non-rhetorical – questions. Bayesian updating might work here.	overall aim of the SDA is the realisation that if you are going to increase your production more in comparison to the IEA predictions for the sector growth, you will have to decrease your carbon intensity more to ensure the sector stays within its budget. The market share of individual companies will change through time and requiring annual updates to the pathway budget, which will require a little bit of work. Bayesian updating sounds like a practical solution for those with the resources.
Figure 1 I assume the measure in panel b is *cumulative* CO2 emissions. I think this should be explicit.	Thank you, we agree with this suggestion. We have added this both in the figure description by adding: “, where Panel a shows carbon intensities and Panel b shows cumulative carbon emissions.” (line 321-322), and added this to the y-axis in Figure 2b.

Reviewer comments, fourth round

Reviewer #1 (Remarks to the Author):

The authors have done a great job at addressing my review comments. I only have a handful of additional minor comments that mostly relate to increasing the clarity of the text:

- Generally: It is not always clear whether you use the "carbon budget" of a company to describe the allocated emissions (as per a global Paris-aligned pathway) in a given year (within a company-level pathway) or to describe the allocated cumulative emission over a time period. Likewise, I have encountered several meanings of the "carbon budget" term elsewhere in the literature and SBTi does not use the term consistently. I therefore encourage you to properly define the "carbon budget" term the first time you use it. Precise language is important here because your proposal for re-aligning pathways serves to adjust the ANNUAL allocated emissions so as to preserve the CUMULATIVE allocated emissions.
- Line 85: Four or five operational requirements? Looks like four in Figure 1.
- Line 101: Consider stating the actual number of companies instead of "a number of major companies".
- Line 118: Consider writing "from which company progress is measured" instead, to make it clear that you are not talking about measurement of global progress.
- Line 142: Consider writing "the start year of the underlying pathway", to be more specific.
- Line 149: Should there be a full stop between "condition" and "as"?
- Table 1: I think you have an error in the analysis of condition #3 for ACA. The 1.5C pathway requires 4.2%/yr, while the well-below 2C pathway requires 2.5%/yr (see SBTi foundations for target setting document).
- Line 172-187: You could perhaps make it clearer that the methods rely on future company-level projections of physical/economic activity (or market share) (not just those variables in the base year).
- Line 214: It is confusing that you write "finally" here, as if you are presenting a third requirement (when it is, in fact, an elaboration of your second requirement).
- Line 235: As I read the SBTi guidance, companies are encouraged to use the latest year for which data is available as base year, provided that that most recent year was not atypical (in terms of emission or activity). This may explain why many companies that recently had their SBTs approved used 2019 as base year (while the use of 2015 or earlier as base year is increasingly rare). You might want to check and maybe integrate that information here.
- Line 296: the line looks more like dark blue than black in the figure?
- Line 297-298: It is a bit confusing that you jump from the single company case (AGL) to a company sample and then back to the single company. Consider moving the information about the company sample further down in the text (around line 352).
- Figure 2b: Consider correcting the arrow from the small window so that it more accurately points at the intersection of the three pathways.
- Line 389: You can maybe cite the Krabbe et al. study again here, which has a great illustration of the effect of grandfathering in its charts.

Reviewer #2 (Remarks to the Author):

I think the authors have done enough for me to recommend publication without further revisions. I still think the problem of revising low-carbon scenarios to take into account e.g. economic and technology shocks is a problem for the method here. For example, Covid-19 has quite significantly changed the IEA pathways in some sectors like aviation. But it isn't a reason to stop the paper being published. I hope the review process has been worthwhile.

Reviewer #1 (Remarks to the Author):	
The authors have done a great job at addressing my review comments. I only have a handful of additional minor comments that mostly relate to increasing the clarity of the text:	We would like to thank the reviewer for continuing to help in improving the manuscript. His or her insights have been invaluable in the revision process!
Generally: It is not always clear whether you use the “carbon budget” of a company to describe the allocated emissions (as per a global Paris-aligned pathway) in a given year (within a company-level pathway) or to describe the allocated cumulative emission over a time period. Likewise, I have encountered several meanings of the “carbon budget” term elsewhere in the literature and SBTi does not use the term consistently. I therefore encourage you to properly define the “carbon budget” term the first time you use it. Precise language is important here because your proposal for re-aligning pathways serves to adjust the ANNUAL allocated emissions so as to preserve the CUMULATIVE allocated emissions.	This is a very good point. We have now specified the following when using the term Carbon Budget for the first time: “Unless specified otherwise, references to carbon budgets in this paper are company-specific cumulative allocated emissions, so that collectively, companies stay within global carbon budgets.” (line 85-87)
Line 85: Four or five operational requirements? Looks like four in Figure 1.	It is four, as we removed one in the prior revision. This paragraph has also been entirely rewritten as per editorial request to focus on what we find. The sentence now reads: “In this work, we first propose two conditions and four operationalisation requirements to evaluate existing methodologies and assess a companies’ Paris Compliance (Figure 1).” (line 72-73)
Line 101: Consider stating the actual number of companies instead of “a number of major companies”.	We now specify the number of companies: “We develop Paris Compliant emissions reduction pathways, and assess the performance using three metrics, for ten companies each from two very different economic sectors; the data-rich Australian electricity industry, and the data-poor global cement production industry.” (line 77-79)
Line 118: Consider writing “from which company progress is measured” instead, to make it clear that you are not talking about measurement of global progress.	We have removed this sentence as we restructured the paragraph to meet editorial requests to focus on findings only.

Line 142: Consider writing “the start year of the underlying pathway”, to be more specific.	Yes, thank you! This has been corrected.
Line 149: Should there be a full stop between “condition” and “as”?	Yes, we have included a full stop and removed “as”
Table 1: I think you have an error in the analysis of condition #3 for ACA. The 1.5C pathway requires 4.2%/yr, while the well-below 2C pathway requires 2.5%/yr (see SBTi foundations for target setting document).	Indeed, this has been corrected.
Line 172-187: You could perhaps make it clearer that the methods rely on future company-level projections of physical/economic activity (or market share) (not just those variables in the base year).	Great point, we have added this in as follows: “It is also important to note that all these methods rely on future company-level projections of physical or economic activity to set targets, which can be different from actual activity.” (line 157-159)
Line 214: It is confusing that you write “finally” here, as if you are presenting a third requirement (when it is, in fact, an elaboration of your second requirement).	Thank you, we agree. We have changed it to “In addition”.
Line 235: As I read the SBTi guidance, companies are encouraged to use the latest year for which data is available as base year, provided that that most recent year was not atypical (in terms of emission or activity). This may explain why many companies that recently had their SBTs approved used 2019 as base year (while the use of 2015 or earlier as base year is increasingly rare). You might want to check and maybe integrate that information here.	Another great addition, thank you! We have rewritten the sentence to include this: “However, the SBTi still allows companies to set a base year beyond 2014, and even if a company sets a target in 2014 (which is very rare, given companies are encouraged to use the latest year for which is data is available), there is no clarity on how to accurately include any deficits it incurs since this base year, or account for changes in their actual vs projected market share, in their target recalculation.” (line 207-211).
Line 296: the line looks more like dark blue than black in the figure?	Yes, thank you for picking this up. We had changed the colour but not all the text. We have corrected this.

Line 297-298: It is a bit confusing that you jump from the single company case (AGL) to a company sample and then back to the single company. Consider moving the information about the company sample further down in the text (around line 352).	We have moved it to your suggested place (now line 417-420).
Figure 2b: Consider correcting the arrow from the small window so that it more accurately points at the intersection of the three pathways.	Thank you – we have done this in the updated figures.
Line 389: You can maybe cite the Krabbe et al. study again here, which has a great illustration of the effect of grandfathering in its charts.	Excellent suggestion, we have added this reference.